# Inference of past demography, dormancy and self-fertilization rates from whole genome sequence data

Thibaut Paul Patrick Sellinger[1]*, Diala Abu Awad[1], Markus Moest[2], Aurélien Tellier[1]

**1** Department of Population Genetics, Technische Universitaet Muenchen, Freising, Germany, **2** Department of Ecology, University of Innsbruck, Innsbruck, Austria

* thibaut.sellinger@tum.de

**Data Availability Statement:** Multihetsep files of Arabidopsis thaliana can be found at: https://github.com/TPPSellinger/Arabidopsis_thaliana_data. Multihetsep files of Daphnia Pulex can be

## Abstract

Several methods based on the Sequential Markovian coalescence (SMC) have been developed that make use of genome sequence data to uncover population demographic history, which is of interest in its own right and is a key requirement to generate a null model for selection tests. While these methods can be applied to all possible kind of species, the underlying assumptions are sexual reproduction in each generation and non-overlapping generations. However, in many plants, invertebrates, fungi and other taxa, those assumptions are often violated due to different ecological and life history traits, such as self-fertilization or long term dormant structures (seed or egg-banking). We develop a novel SMC-based method to infer 1) the rates/parameters of dormancy and of self-fertilization, and 2) the populations' past demographic history. Using simulated data sets, we demonstrate the accuracy of our method for a wide range of demographic scenarios and for sequence lengths from one to 30 Mb using four sampled genomes. Finally, we apply our method to a Swedish and a German population of *Arabidopsis thaliana* demonstrating a selfing rate of *ca.* 0.87 and the absence of any detectable seed-bank. In contrast, we show that the water flea *Daphnia pulex* exhibits a long lived egg-bank of three to 18 generations. In conclusion, we here present a novel method to infer accurate demographies and life-history traits for species with selfing and/or seed/egg-banks. Finally, we provide recommendations for the use of SMC-based methods for non-model organisms, highlighting the importance of the per site and the effective ratios of recombination over mutation.

## Author summary

With the rapid advancement of sequencing technologies it has become feasible to use genome sequence data from several individuals per population/species to learn about the past history of populations, such as changes in population size, timing of colonization of a given habitat/continent or past migration events. However, all statistical methods to date rely on a mathematical model built upon characteristics of hominid species.

found at: https://github.com/TPPSellinger/
Daphnia_pulex_data.

**Funding:** This work was funded by 809_7/1
Deutsche Forschungsgemeinschaft (https://www.
dfg.de/) to AT, TUM fellowship (https://www.tum.
de/en/research/postdocs/research-opportunities-
week/tum-university-foundation-fellowship/) to
DAA, and The Austrian Science Fund (FWF):
P29667-B25 (https://m.fwf.ac.at/en/research-
funding/fwf-programmes/) to MM. The funders
had no role in study design, data collection and
analysis, decision to publish, or preparation of the
manuscript.

**Competing interests:** The authors have declared
that no competing interests exist.

Namely, this model ignores specific ecological traits common to many plant, inverte-
brate, or fungal species such as self-fertilization and dormancy. The latter is defined as
the ability of plant seeds or invertebrates to remain dormant for long periods of time in
the soil or sediments. Here we develop a new statistical method which uses several whole
genome sequence data per population/species to simultaneously infer 1) the rates of dor-
mancy and of self-fertilization, and 2) past demographic history. We demonstrate the
accuracy of our method using extensive simulations. We also apply our method to *Arabi-
dopsis thaliana* confirming a high rate of selfing and the absence of long term seed dor-
mancy, and to *Daphnia Pulex* demonstrating a long lived egg bank. We discuss the wide
applicability of our method to draw accurate inference of past evolution for non-homi-
nid species.

## Introduction

Genomes, and especially genetic polymorphisms, are shaped by molecular forces, such as
mutation and recombination, but also ecological forces intrinsic to, or independent of, the
biology of the species [1]. Polymorphism data therefore contain a plethora of information that
goes beyond the physiological functions encoded therein. Recent advances in sequencing tech-
nologies enable us to obtain whole genome data for many individuals across several popula-
tions, even for non-model species [2–5]. In particular, inferring demography is of interest in
its own right as it allow us to understand the history of existing and/or extinct species (popula-
tion expansion, colonization of new habitats, past bottlenecks) [5–7]. Inferring demography is
also necessary to generate null models for outlier scans, *e.g.* scanning genomes for genes under
selection. Indeed, inaccurate demographies may strongly bias the outcome of such scans [8]. It
is now common practice to simulate the past demography of a population as a null model in
order to define thresholds for selection scan methods. Therefore, an accurate demographic
inference should yield more reliable selection results [8, 9]. To this aim, new models and meth-
ods have been developed to extract previously unavailable information from whole genome
sequence data [10–14]. Inference is based on modeling single nucleotide polymorphism
(SNPs) along the genome across individuals, the density of which results from the interplay
between mutation, time to common ancestors and recombination. The common denominator
in all these methods is their reliance on the per site ratio recombination ($r$) and the mutation
($\mu$) rate of the species ($\frac{r}{\mu}$), or, more precisely, on its effective value $\frac{\rho}{\theta}$. We classically define $\rho$ as
the effective recombination rate and $\theta$ as the effective mutation rate. We note that so far, appli-
cations of these approaches have considered these ratios as interchangeable [10–12], which is a
strong assumption and may be violated in some species that do not fulfill the assumptions of
the classic Wright-Fisher diploid model with two sexes (*e.g.* equal sex-ratio, sexual reproduc-
tion at each generation and no overlap of generations). In humans and mammals, as $\rho = 4N_e r$
and $\theta = 4N_e\mu$ ($N_e$ being the effective population size), we indeed find $\frac{r}{\mu} = \frac{\rho}{\theta}$. Yet, even in this
case, biases arise if $\frac{\rho}{\theta} > 1$ [10]. In such cases, the number of mutations is not sufficient to detect
all recombination events. The model is therefore no longer able to correctly estimate the
Ancestral Recombination Graph, *i.e* the superposition of coalescence trees at different posi-
tions on the genome to display genealogies of sequences in the presence of recombination.
Generally, it becomes necessary to extend existing approaches to account for characteristics
and traits of species that can influence $\rho$ or $\theta$, and thus define when these methods can be accu-
rately applied. It is additionally of interest to assess the accuracy of such methods for various
values of the ratio $\frac{\rho}{\theta}$.

Current methods rely on the Sequentially Markovian coalescence (SMC) [15, 16] to account for the linear structure of genome sequences. SMC models the genealogy of a sample along a chromosome. First, a genealogy is built under the neutral coalescence and, in a second step, recombination and linkage disequilibrium are incorporated using a Poisson process [17, 18]. By applying Hidden Markov Models it therefore becomes possible to calculate the probabilities of whole genome sequence data and infer the most likely values of the model parameters. These approaches can thus infer 1) the changes in population size ($\chi_t$, where $\chi_t = \frac{N_t}{N_e}$, $N_e$ and $N_t$ being the effective population size and the current population size at time $t$, respectively) by inferring any variation of the coalescence rate in time, and 2) the ratio of effective recombination over the effective mutation rate $\frac{\rho}{\theta}$. From this ratio, the recombination rate can be estimated if the mutation rate is known (assuming $\frac{\rho}{\theta} = \frac{r}{\mu}$).

The described methods have been almost exclusively built to be applied to hominid data, therefore rely on several assumptions that are violated in many species (and most likely also in hominids): non-overlapping generations, equal sex ratio, sexual reproduction through random mating. Indeed, with the rise of next-generation sequencing technology, these methods are now frequently applied to whole genome sequences of species with characteristics that greatly differ from humans [3, 4, 19, 20]. In many species (*e.g.* plants, invertebrates) life-history strategies, such as mating systems or offspring production, influence the relationship between $r$ and $\rho$ and $\mu$ and $\theta$ [1]. If these effects are not accounted for, inferences using these methods may be biased and lead to misinterpretation of the results.

Two very common features in plant and invertebrate species are the maintenance of offspring as seed- or egg-banks [21–23] and self-fertilization [24]. Indeed, as a consequence of environmental fluctuations, species can develop bet-hedging strategies such as seed-banking [25–27]. This strategy increases the observed diversity [28, 29] and affects the rate of selection and neutral genomic evolution [30, 31]. Due to the discrepancy between census ($N_{cs}$) and effective population size ($N_e$) caused by seed-banks [28, 32], we expect that $\frac{\rho}{\theta} \neq \frac{r}{\mu}$. Seed-banks can therefore strongly bias demographic inference if ignored [33]. Self-fertilization, on the other hand, decreases the effective population size. This reproductive strategy has evolved many times independently, and is one of the most common evolutionary transitions observed in flowering plants [34]. The main consequence of this mating system is an increased homozygosity, which directly results in a decreased effective recombination rate ($\rho$) compared to the molecular recombination rate $r$ (since recombination events between two homozygous haplotypes are invisible), as well as a reduction in genetic diversity [35]. Due to their contradictory effects on the effective population size, the simultaneous occurrence of these traits (dormancy and self-fertilization) may in fact be missed, and extensions of inference methods to account for them could not only allow for more accurate inferences of parameters and demographic histories of species with these traits, but could also provide a means with which to detect their respective rates.

To account for self-fertilization and seed-banks (or egg-banks) we develop a modified version of PSMC' [12], named ecological Sequentially Markovian Coalescent (eSCM). PSMC' refers to the MSMC using only two haplotypes [12], which is slightly different from the original PSMC [11]. Our model uses the deviation between the ratios $\frac{\rho}{\theta}$ and $\frac{r}{\mu}$ to infer self-fertilization and the existence of seed-banks. However, confounding effects arise when estimating both simultaneously. We first apply eSMC to simulated data to demonstrate its accuracy and then to genome sequence data of a plant, *Arabidopsis thaliana*, and an invertebrate species, *Daphnia pulex*. In these species self-fertilization and/or seed/egg-banks have been observed or

suspected. *A. thaliana* presents a very high self-fertilization rate of 99% [36] and it has been suggested that Scandinavian populations may have evolved seed-banks in Sweden [37] and Norway [38]. *D. pulex* exhibits cyclical parthenogenesis, *i.e.* a cyclical alternation of phases with asexual and sexual reproduction and is known to have a dormant eggs produced through sexual reproduction [4, 39]. These resting eggs can potentially build up an egg-bank in the lake sediment as observed in many *Daphnia* species [21, 40]. The aim of our study is thus three-fold. First, we present a method based on the SMC using polymorphism data to infer the germination and/or self-fertilization rates jointly with the past demographic history. Second, we study the effect of variable ratios of $\frac{\rho}{\theta}$ (and $\frac{r}{\mu}$) on the accuracy of estimates of past demography. Third, we apply our method to existing datasets from *Arabidopsis thaliana* and *Daphnia pulex* which have well documented high self-fertilization rates and egg-banks, respectively. We find a strong signature of self-fertilization in *Arabidopsis thaliana* and a strong signature of egg-banks in *Daphnia pulex*. We find that self-fertilization has little effect on the inference of the demographic history, whereas neglecting seed-banks can strongly affect the inferred population size.

## Overview of the model

### The coalescence with seed-bank and self-fertilization

We model population seed-banks using the same hypotheses described in [41]. Under these assumptions, seed-banking can be accounted for by rescaling the coalescence rate by $\beta^2$, where $\beta$ ($0 \leq \beta \leq 1$) is the germination rate, or the expected germination probability in each generation ($\beta = 1$ implying that there is no seed-bank). The probability that two lineages find a common ancestor in the active population is slowed by a factor $\beta \times \beta$ when looking backward in time. Hence, the expected coalescence times are increased by a factor $\frac{1}{\beta^2}$. Assuming mutations can arise during the dormant stage at the same rate as in the active population, we expect to have $\frac{1}{\beta^2}$ more mutations [25, 28, 30]. As recombination only occurs in the active population and concerns only one lineage backward in time, it is rescaled by $\beta$ [31]. Because coalescence times are $\frac{1}{\beta^2}$ longer, we obtain (scaled in units of 4N):

$$\rho = \frac{\beta r}{\beta^2} = \frac{r}{\beta} \ \text{ and } \ \theta = \frac{\mu}{\beta^2}, \ \text{ so that } \ \frac{\rho}{\theta} = \frac{\beta^2 r}{\mu \beta} = \frac{\beta r}{\mu} \tag{1}$$

To model self-fertilization, we adopted the island model described in [42], where $\sigma$ ($0 \leq \sigma \leq 1$) represents the proportion of offspring produced through self-fertilization (if $\sigma = 1$ all individuals are produced through self-fertilization). As a consequence, the coalescence rate is increased by a factor $\frac{2}{2-\sigma}$ [43] and the recombination rate is decreased by a factor $\frac{2-2\sigma}{2-\sigma}$ [42] since recombination events in homozygous individual are invisible. In the case of self-fertilization, we thus find (scaled in units of 4N):

$$\rho = \frac{2(1-\sigma)r}{(2-\sigma)} \frac{(2-\sigma)}{2} = (1-\sigma)r \ \text{ and } \ \theta = \frac{\mu(2-\sigma)}{2}, \tag{2}$$

$$\text{so that } \ \frac{\rho}{\theta} = \frac{r2(1-\sigma)}{\mu(2-\sigma)} \tag{3}$$

To simultaneously model seed-banking and self-fertilization we assume their effects to be independent and that there is no correlation between dormancy and the rate of self-fertilization. Under this assumption we can simply multiply their effects as in [29], giving the

relationship $\frac{\rho}{\theta} = \frac{2(1-\sigma)\beta r}{(2-\sigma)\mu}$. We therefore have a confounding effect between self-fertilization and seed-banking when observing the recombination and mutation ratio $\frac{\rho}{\theta}$. Because of their opposing effects on the effective population size (seed dormancy increasing it, and self-fertilization decreasing it), the effects of these traits can be compensated by one another. As consequence, in our model seed-banking is mathematically equivalent to self-fertilization with higher effective population size (see the Discussion for ruling out this effect in practice).

## ecological Sequentially Markovian Coalescent (eSMC)

eSMC is an extension to the PSMC' algorithm [12] and is therefore a Hidden Markov Model (HMM) along two haplotypes. It adds the possibility to account for and estimate germination rates $\beta$ (seed and egg-banks), as well as self-fertilization rates $\sigma$, while inferring the demographic history of a population/species ($\chi_t$). Our method is hence, in effect, a rescaled PSMC'. The input file contains all pairwise comparisons of genome sequences; at each position if the two nucleotides on each sequence are the same, this is indicated by 0, otherwise by 1.

The general idea is that mutations are more likely to arise if the coalescence time is long between the two lineages, therefore, as in PSMC', the hidden states are coalescence times (where time is discretized). In SMC, the coalescent model is the classic $n$-Kingman coalescent with sample of size two and based on the assumptions of the classic Wright-Fisher model. However, because seed-banks and self-fertilization scale the coalescent rate, we re-scale the hidden states by $\frac{(2-\sigma)}{2\beta^2}$, where $\sigma$ and $\beta$ are respectively the self-fertilization rate and the germination rate. Variations in the coalescence times, or the hidden states, along the sequence, are caused by recombination events. In order to account for the effects of seed-banks and self-fertilization on the observed variations, the recombination rate must be re-scaled by $\frac{2(1-\sigma)\beta}{(2-\sigma)}$. A detailed description of the model can be found in S1 of the Appendix and a list of symbols and parameters in Table 1.

We use a Baum-Welch Algorithm to find the estimates of our parameters (the effective recombination rate $\rho$, the germination rate $\beta$, the self-fertilization rate $\sigma$, and changes in population size $\chi_t$ [10, 12]. Symbol definitions are summarized in Table 1. The Baum-Welch Algorithm aims to maximize an objective function ($L$) analogous to the likelihood [12]. In addition, unlike in PSMC', MSMC and MSMC2 in which the objective function is calculated only every 1 kb, our implementation calculates the objective function at every position of the sequence.

To increase the model's accuracy, several analyses are jointly run (similar to what is done in [44]). As all individuals are sampled from a given population, we compute a composite objective function $CL$, which is the product of all the objective functions (calculated for each analyses). Assuming we have $n$ diploid individuals (equivalent to $2n$ sequences), we perform $\binom{2n}{2}$

**Table 1. Symbol table.**

| Symbol | Meaning |
|---|---|
| r | Molecular Recombination rate per nucleotide per generation |
| $\mu$ | Molecular Mutation rate per nucleotide per generation |
| $\rho$ | Effective recombination rate per nucleotide per generation |
| $\theta$ | Effective mutation rate per nucleotide per generation |
| $\beta$ | Germination rate |
| $\sigma$ | Self-fertilization rate |
| $\chi_t$ | Population size scaling vector (i.e $N_t = \chi_t N_0$) |

analyses (number of all pairwise analyses) on the $2n$ haplotypes (requiring phasing). If sequences cannot be phased, we perform $n$ analyses, one for each diploid individual. It can also occur that individuals are higly homozygote (e.g. *A. thaliana*), in that case individual can be considered haploid [19]. Therefore if we have $n$ higly homozygote individuals, we have $n$ sequences and can perform $\binom{n}{2}$ analyses (since no phasing is required). The algorithm is run to infer $\beta$, $\sigma$ and $\chi_t$ by fixing or simultaneously inferring the ratio $\frac{\rho}{\theta}$ given an initial value of this ratio.

Although mathematically equivalent to a rescaled PSMC', our model is the first to integrate ecological knowledge and infer ecologically relevant parameters. Unlike MSMC, PSMC' and MSMC2, in eSMC the likelihood is optimized under constraints. This means that estimated values are bounded by prior knowledge that is set by the user (this includes selfing rates, the existence of seed banks and also population sizes). This feature prevents any estimation which would make little sense from the biological point of view (e.g. a self-fertilization rate below 0 or above 1) or from an ecological point of view (e.g. excessively low or high population size). From a methodological point of view, eSMC is tuned to outperform previous methods when dealing with small datasets and genomes of limited size from non-model species, such as fungi, protozoans, insects or plants. Furthermore, eSMC differs in its implementation from previous methods in several key aspects: 1) A new analysis scheme over all haplotype pairwise comparisons has been implemented, 2) We include the possibility to optimize the likelihood under constraint, 3) The emission matrix has been modified compared to PSMC' and MSMC, and 4) A different implementation of the Baum-Welch algorithm has been developed.

Our model is implemented in the R-package eSMC and can be downloaded from our GitHub repository (https://github.com/TPPSellinger/eSMC, note that the R package devtools is required for the installation). The algorithm uses the same input file format as MSMC or PSMC'. In comparison with other similar models, our implementation in R is slightly slower (examples of calculation times can be found in Table 2), but still allows whole genome sequence analysis of several individuals in a reasonable amount of time. Furthermore, we added a feature to eSMC allowing the simulation of pseudo-observed data based on the obtained results. These pseudo-observed data are then analyzed by eSMC to asses the robustness of the results.

## Results

We first study the theoretical accuracy and properties of our method on sequence data simulated under different scenarios. We then analyze real sequence data from two European populations of *Arabidopsis thaliana*: one from Tübingen, Germany, where there is no seed-bank, and one from Sweden, where seed-banking is suspected, while accounting for self-fertilization. We also analyze data from *Daphnia pulex*, for which egg-banks are known to be a prominent biological feature.

**Table 2. Calculation times of eSMC on simulated data under the "saw-tooth" demographic scenario with mutation and recombination rate set to $2.5 \times 10^{-8}$ per generation per bp.** Results are in minutes given the sequence length and the number of haplotypes.

| Sequence length (Mb) | 2 Haplotypes | 4 Haplotypes | 10 Haplotypes |
|---|---|---|---|
| 1 Mb | 12 | 13 | 14 |
| 10 Mb | 14 | 16 | 31 |
| 30 Mb | 19 | 25 | 75 |

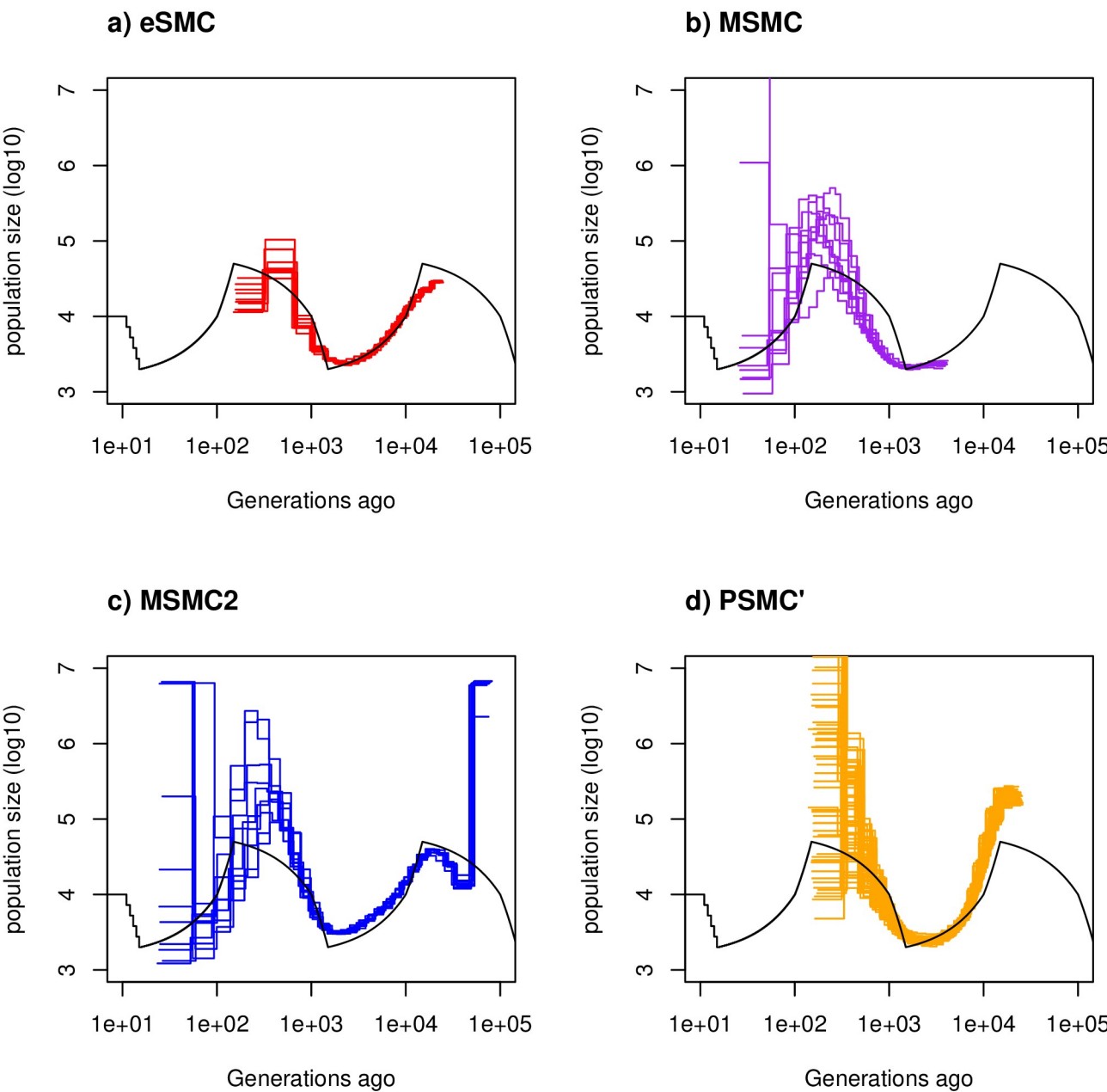

**Fig 1. Estimated demographic history with no selfing or seed banking.** Estimated demographic history using four simulated sequences of 30 Mb under a saw-tooth scenario with 10 replicates. Mutation and recombination rates (respectively $\mu$ and $r$) are set to $2.5 \times 10^{-8}$ per generation per bp. Therefore $\frac{r}{\mu} = \frac{\rho}{\theta} = 1$. The simulated demographic history is represented in black. a) Demographic history estimated by eSMC (red). b) Demographic history estimated by MSMC (purple). c) Demographic history estimated by MSMC2 (blue). d) Demographic history estimated by PSMC' (orange).

## Simulation results

**Convergence property in the absence of seed-banks and self-fertilization.** We start by analyzing 4 sequences of 30 Mb, without seed-banks or self-fertilization, from a population simulated under a "saw-tooth" scenario (repetitions of an expansion followed by a decrease, similar to those used in [10, 12]). In these simulations the ratio $\frac{r}{\mu}$ is allowed to be inferred freely. From Fig 1, we see that all of the methods tested with this scenario (eSMC, MSMC,

**Table 3. Averaged estimated values for the recombination over mutation ratio $\frac{\rho}{\theta}$ over ten repetitions.** The coefficient of variation is indicated in brackets.

| Scenario | Sequence length | real $\frac{\rho}{\theta}$ | eSMC $\frac{\rho}{\theta}$ | PSMC' $\frac{\rho}{\theta}$ | MSMC $\frac{\rho}{\theta}$ | MSMC2 $\frac{\rho}{\theta}$ |
|---|---|---|---|---|---|---|
| "Saw-tooth" | 30 Mb | 1 | 0.90 (0.028) | 0.93 (0.051) | 1.28 (0.050) | 0.38 (0.088) |
| "Saw-tooth" | 10 Mb | 1 | 0.94 (0.087) | 0.96 (0.11) | 1.30 (0.13) | 0.39 (0.21) |
| "Saw-tooth" | 10 Mb | 5 | 2.21 (0.040) | 2.06 (0.050) | 2.77 (0.049) | 1.47 (0.062) |
| "Saw-tooth" | 1 Mb | 1 | 0.96 (0.14) | 1.00 (0.25) | 1.31 (0.28) | 0.86 (0.2) |
| Constant | 10 Mb | 1 | 0.96 (0.035) | 0.96 (0.056) | 1.06 (0.10) | 0.81 (0.089) |
| Bottleneck | 10 Mb | 1 | 1.09 (0.026) | 1.09 (0.048) | 1.02 (0.107) | 1.01 (0.033) |
| Expansion | 10 Mb | 1 | 1.05 (0.038) | 1.06 (0.059) | 1.07 (0.120) | 0.98 (0.074) |
| Decrease | 10 Mb | 1 | 0.89 (0.028) | 0.90 (0.047) | 1.1 (0.186) | 0.69 (0.055) |

The "saw-tooth" estimations are seen in Fig 1, and the other scenarios are shown in S3–S6 Figs.

MSMC2 (version 2.1.1) and PSMC') show an accurate fit and small variability of the estimated demographic histories. As expected from previous results, with a population size of approximately $10^4$, inference of very recent ($< 10^2$ generations) and very old events ($> 10^5$ generations) cannot be recovered due to limits of the coalescent resolution. Estimates of $\frac{r}{\mu}$ by eSMC, PSMC' and MSMC are accurate (Table 3), whereas MSMC2 exhibits a strong bias, generally underestimating the recombination rate in this scenario. Though we find that eSMC displays less variability compared to the other methods, it is important to note that these methods do not all share the same time windows to estimate the demographic history, and this cannot easily be accounted for. eSMC and PSMC' use the same time window, whereas MSMC uses a time window six times more recent than that of the previous two (as a consequences of using four haplotypes), and MSMC2 uses the largest time window, which is the same as that used in PSMC [11].

The main consequence of these differences in time windows is that population size is not discretized in the same way in all models, *i.e.* there are differences in default number of estimated demographic parameters. In order to evaluate how this affects the variance of each method, we use shorter sequences of 10 MB, using the default discretization of population size parameters of each method (similar to Fig 1). eSMC and PSMC' display convergence properties similar to those with 30 Mb sequences, but MSMC and MSMC2 result in higher variances and less accurate estimates of the demographic histories (S1 Fig). An analysis run using the same discretization of the population size parameters as eSMC for all the methods (S2 Fig) shows that, while PSMC', MSMC and MSMC2 present lower variance and better estimates than in S1 Fig, eSMC still presents the lowest variance. As eSMC has good convergence properties with 10 Mb sequences all the following analyses are carried out with this sequence length, as it is representative of the genome lengths found in many datasets. In addition, since the longest scaffolds from *Daphnia pulex* are in fact very short (length shorter than 2 Mb) we also tested each method with 1 Mb long sequences. As can be seen in S3 Fig, though every method tested exhibits higher variances when using such short sequences, eSMC exhibits a more accurate demographic history and the lowest variance.

For all the demographic scenarios tested (constant population size, bottleneck, expansion, decrease), eSMC always presents the lowest variance, and maintains good convergence properties, although it exhibits some bias in the bottleneck scenario (S4 Fig). The bottleneck is estimated as a smooth curve and not a sharp change in population size. Running PSMC', MSMC and MSMC2 with their default values, with the exception of setting the initial value of $\frac{\rho}{\theta}$ to 1,

we find that PSMC' reveals convergence properties similar to eSMC (S5 Fig), MSMC strongly overestimates population sizes for recent times, but is able to estimate a more recent demography (S6 Fig), and, lastly, MSMC2 shows biased results, with an overestimation of population sizes in the far past, and fails to detect expansions and constant demography (see S7 Fig). All estimated recombination over mutation ratios can be found in Table 3.

We now assume $\frac{\rho}{\theta} = \frac{r}{\mu} = 5$, with the mutation and recombination rate respectively set to $2.5 \times 10^{-8}$ and $1.25 \times 10^{-7}$ per generation per nucleotide. Note that fixing $\frac{\rho}{\theta} = 5$ means that we estimate the demographic history conditioned on this fixed ratio (which we know to be the true one). Hence, the ratio $\frac{r}{\mu}$ is no longer free to be inferred but fixed. All methods provide strongly biased estimates (S8 Fig) and the estimated demographic histories are very flat. Increasing the ratio $\frac{\rho}{\theta}$ to 100 (or in this case $\frac{r}{\mu} = 100$) further, the estimated demographic history tends to a constant population size (S9 Fig). If we now allow the methods to simultaneously estimate demography and the ratio $\frac{\rho}{\theta}$ (the ratio $\frac{r}{\mu}$ is now free to be inferred), using $\frac{r}{\mu}$ (the real value of the molecular ratio) as an input parameter (or initial value) for the estimated $\frac{\rho}{\theta}$, the results exhibit smaller but similar biases as in S8 Fig (see S10 Fig). However if the initial value for $\frac{\rho}{\theta}$ is set to 1 the results exhibit almost no bias (S11 Fig). In this case, the demographic history is accurately estimated although the estimated ratio $\frac{\rho}{\theta}$ is smaller than $\frac{r}{\mu}$ (see Table 3). While of general importance, these results have not been highlighted in previous works and we discuss their relevance later on. In the following analyses, we launch eSMC, PSMC', MSMC and MSMC2 to estimate the demographic history and the ratio $\frac{\rho}{\theta}$ kept free to be inferred using $\frac{\rho}{\theta} = 1$ as the initial value.

**Convergence property with dormancy (seed- or egg-banks).** Using eSMC on sequences simulated under the "saw-tooth" scenario in the presence of seed-banks (mutation and recombination rates are set to $2.5 \times 10^{-8}$ per generation per bp, Fig 2), we obtain an accurate estimation of the demography ($\chi_t$) and of the germination rates ($\beta$). Under four germination rates $\beta$ with values 1 (no seed-bank), 0.5 (two-year seed-bank), 0.2 (long-lived five-year seed-bank) and 0.1 (long-lived ten-year seed-bank), we respectively estimate an average germination rate of 0.92, 0.53, 0.22 and 0.09. As seed-banks affect the time window of the estimated demography, more ancient events can be inferred when $\beta < 1$ [33]. In models where seed-banks cannot be accounted for (PSMC', MSMC, MSMC2), census population size is strongly overestimated when $\beta < 1$. When seed-banks are long lived (*e.g.* $\beta = 0.1$, giving a mean dormancy of ten generations), eSMC slightly underestimates $\beta$ and the population size. This is because in presence of strong seed-banks, coalescence times increase, which can lead to violation of the infinite site model. Therefore when the molecular mutation and recombination are set to $5 \times 10^{-9}$ per generation per bp, better fits are obtained (S12 Fig).

For simpler demographic scenarios (constant population size, bottleneck, expansion and decrease, see S13 Fig) and $\mu = r = 2.5 \times 10^{-8}$ per generation per bp, the germination rate and the demographic histories estimated by eSMC are accurate for most of the demographic scenarios considered, except in the case of a bottleneck scenario (as expected from previous results). In presence of strong seed-banks ($\beta = 0.2$ or 0.1) there are biases in estimations of the far past. Once again, this tendency disappears when the molecular mutation and recombination rates per site are lowered so as not to violate the infinite site model ($\mu$ and $r = 5 \times 10^{-9}$ per generation per bp, see S14 Fig).

**Convergence property with self-fertilization.** Under the "saw-tooth" scenario with different rates of self-fertilization $\sigma$, with mutation and recombination rates set to $2.5 \times 10^{-8}$ per generation per bp $\left(\frac{r}{\mu} = 1\right)$, for four different self-fertilization rates $\sigma = 0$ (no self-fertilization), 0.5 (50% selfing), 0.8 (80% selfing) and 0.9 (90% selfing), we estimate the self-fertilization rate

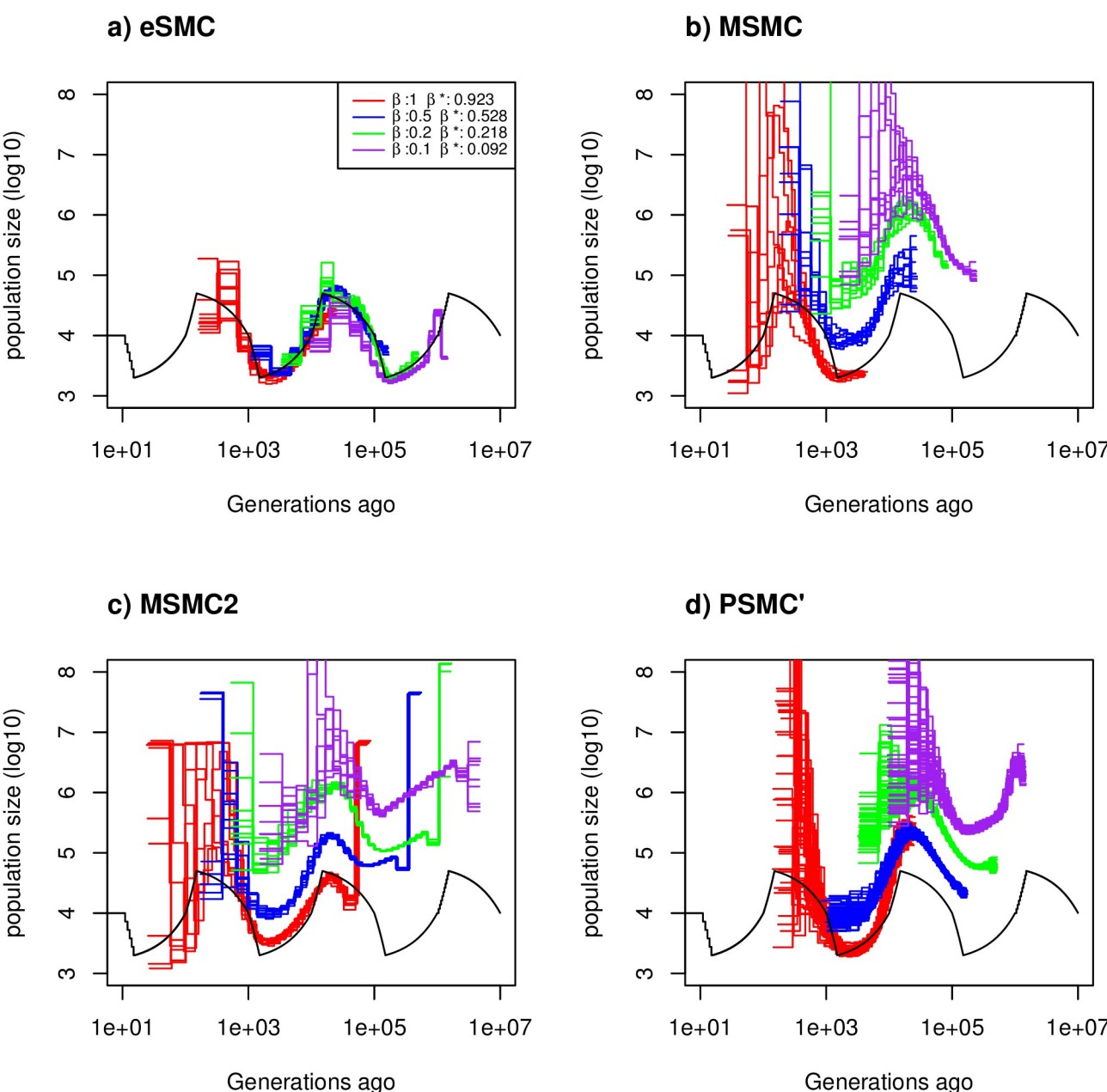

**Fig 2. Estimated demographic history with seed banking.** Estimated demographic history using four simulated sequences of 10 Mb and ten replicates under a saw-tooth demographic scenario (black). The mutation and recombination rates are set to $2.5 \times 10^{-8}$ per generation per bp. Therefore $\frac{r}{\mu} = 1$. We simulate under four different germination rates $\beta = 1$ (red), 0.5 (blue), 0.2 (green) and 0.1 (purple), hence we respectively have $\frac{\rho}{\theta} = 1$, = 0.5, 0.2 and 0.1. The demographic history is estimated using a) eSMC, b) MSMC, c) MSMC2 and d) PSMC'. $\beta^*$ represents the estimated germination rate by eSMC.

respectively at 0.19, 0.5, 0.77 and 0.87 (Fig 3). In addition, we find that eSMC accurately estimates demography, while MSMC, MSMC2 and PSMC' exhibit a small bias in the estimation of the demographic history. Neglecting self-fertilization therefore seems to be of smaller consequence than neglecting dormancy (see above), as self-fertilization has a very small impact on the the inferred demographic history. Variance in the estimations increase for higher rates of $\sigma$. When the mutation rate is set to $2.5 \times 10^{-8}$ per generation per bp and the recombination

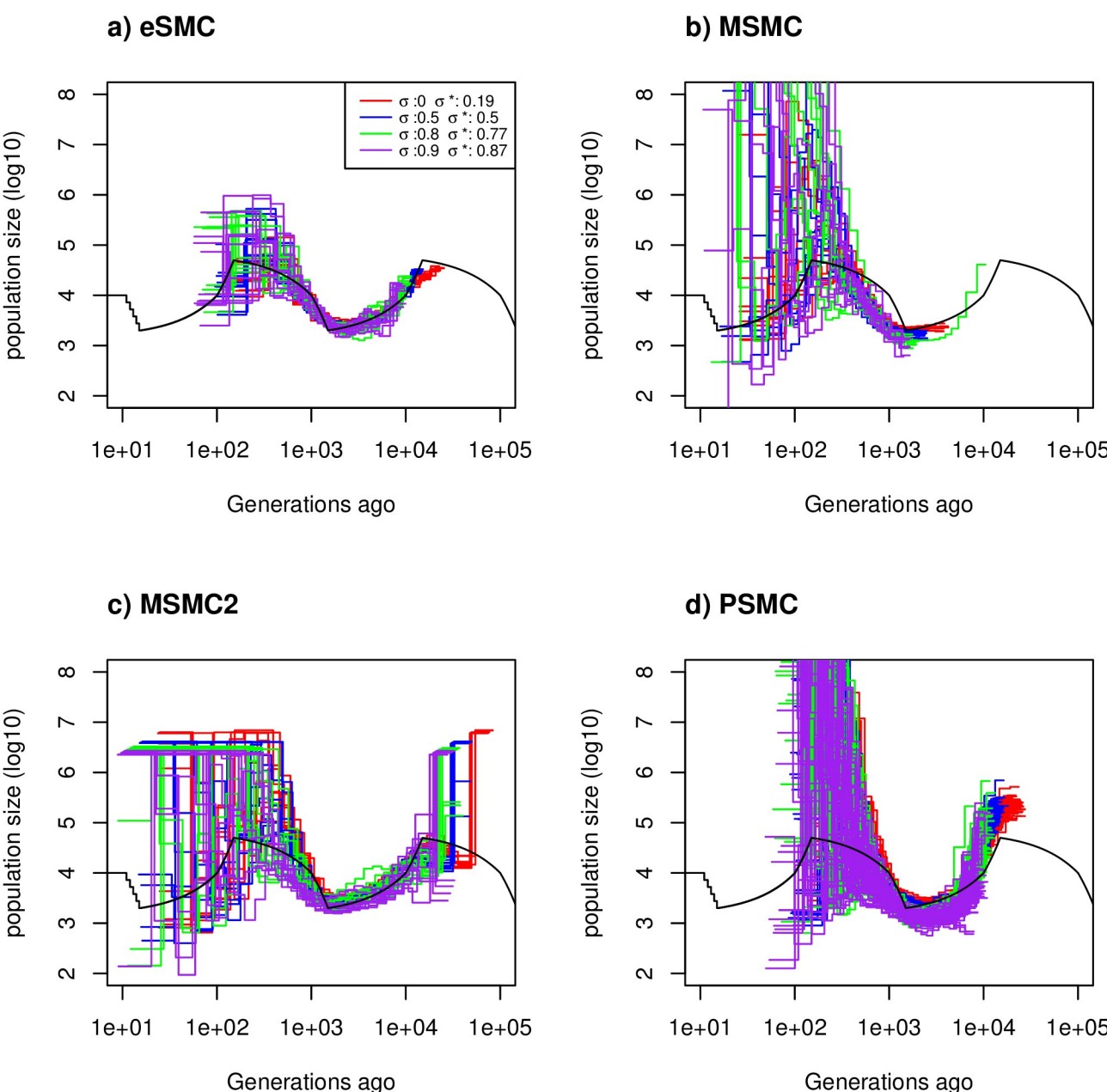

**Fig 3. Estimated demographic history with selfing.** Estimated demographic history using four simulated sequences of 10 Mb and ten replicates under a saw-tooth demographic scenario (black). The mutation and recombination rates are set to $2.5 \times 10^{-8}$ per generation per bp, and simulations were run for four different self-fertilization rates ($\sigma = 0$ (red), 0.5 (blue), 0.8 (green) and 0.9 (purple)), and as $\frac{r}{\mu} = 1$, this gives $\frac{\rho}{\theta} = 1$, 0.5,0.2 and 0.1 respectively. The demographic history is estimated using a) eSMC, b) MSMC, c) MSMC2 and d) PSMC'. $\sigma^*$ represents the self-fertilization rate estimated by eSMC.

rate to $1.25 \times 10^{-7}$ per generation per nucleotide $\left(\frac{r}{\mu} = 5\right)$, the self-fertilization rate is overestimated for small values of $\sigma$ (S15 Fig), but well estimated for higher values of $\sigma$. The estimation of the demographic history remains accurate, however, though slightly biased for small values of self-fertilization. The other methods tested (PSMC', MSMC, MSMC2) present stronger biases in the estimated demographic history.

In the simpler demographic scenarios tested (S16 Fig), the rate of self-fertilization is estimated fairly well, though there is an impact of the considered demographic scenario. However, in absence of self-fertilization, eSMC infers a residual rate of self-fertilization (below 0.2). While, the demographic history is accurately estimated.

**Convergence property with both dormancy and self-fertilization.** Here we test different combinations of seed/egg-banks and self-fertilization rates that result in the same ratio $\frac{\rho}{\theta} = 0.15$, with $\frac{r}{\mu} = 1$ (setting $\mu = r = 2.5 \times 10^{-8}$ per generation per bp). Self-fertilization and dormancy have opposing effects on the coalescence rate, and thus cannot be simultaneously estimated from whole-genome data alone (Fig 4). These two rates are indeed simultaneously non-identifiable. Without any prior knowledge (blue in Fig 4), it is not possible for eSMC to estimate the correct set of parameters. However, this shortcoming can be corrected to some extent by setting general "ecological" priors for either $\beta$ or $\sigma$ (e.g. $0 \le \beta \le 0.5$ or $0.5 \le \sigma \le 1$). In this case eSMC is able to infer a demographic history of the correct shape, but slightly shifted away from the true values of population size and time. eSMC tends to overestimate the values of $\beta$ and $\sigma$, a consequence of which is the overestimation of the census population size (see Fig 4). However, while integrating prior knowledge on both parameters does not solve the non-identifiability issue, it does reduce the inferred range of values. Hence, including priors on both rates reduces the parameter space for which the confounding effect of joint estimation occurs. This is shown in S17 Fig in a plot showing all possible estimations of coupled variables of $\beta$ and $\sigma$ for a given parameter set. We also test how recombination can influence the output of these models, notably by taking a higher recombination rate $(1.667 \times 10^{-7}$ per site per generation), more representative of the high recombination to mutation ratio observed in some species (notably *D. pulex* and *A. thaliana* [4, 45]). This gives $\frac{r}{\mu} = 6.667$ and $\frac{\rho}{\theta} = 1$, parameters for which the variance of the demographic history is smaller and the estimation of self-fertilization and germination parameters remain unchanged (S18 Fig). All the possible estimated combinations of $\beta$ and $\sigma$, given different sets of priors for this recombination rate and ratio $\frac{r}{\mu}$ are given in S19 Fig (results similar to those for $\frac{r}{\mu} = 1$ in S17 Fig are observed).

## Inferring self-fertilization, seed-banks and demography in *Arabidopsis thaliana*

Using 12 individual whole genome sequence data obtained from two accessions of *A. thaliana* (one from Sweden and the other from Germany), we inferred the demography of each population using eSMC, PSMC', MSMC and MSMC2 (S20 Fig). When ignoring self-fertilization, both populations have a common demographic history, similarly inferred by the different methods, except for MSMC, whose results exhibit a higher variance for the Swedish population. Furthermore, we observe a non-negligible deviation between the recombination rate estimated using these inference methods ($\frac{\rho}{\theta} < 1$) and what has been obtained using experimental approaches ($\frac{r}{\mu} = 5$) [45]. When accounting only for self-fertilization (hence imposing $\beta = 1$), eSMC estimates a high self-fertilization rate averaged at $\sigma = 0.86$ in the German population and 0.87 in the Swedish one. These rates are not as high as what has been recorded previously [36, 46]. When running analyses per chromosome, we found no significant chromosome effect on these estimations. When simultaneously estimating $\beta$, $\sigma$ and the population size, we find a slightly lower $\sigma = 0.84$ in the German population and 0.86 in the Swedish one (Fig 5). Here, eSMC estimates a germination rate $\beta$ higher than 0.9 in both populations, implying that there is no long-term seed-bank in either of them.

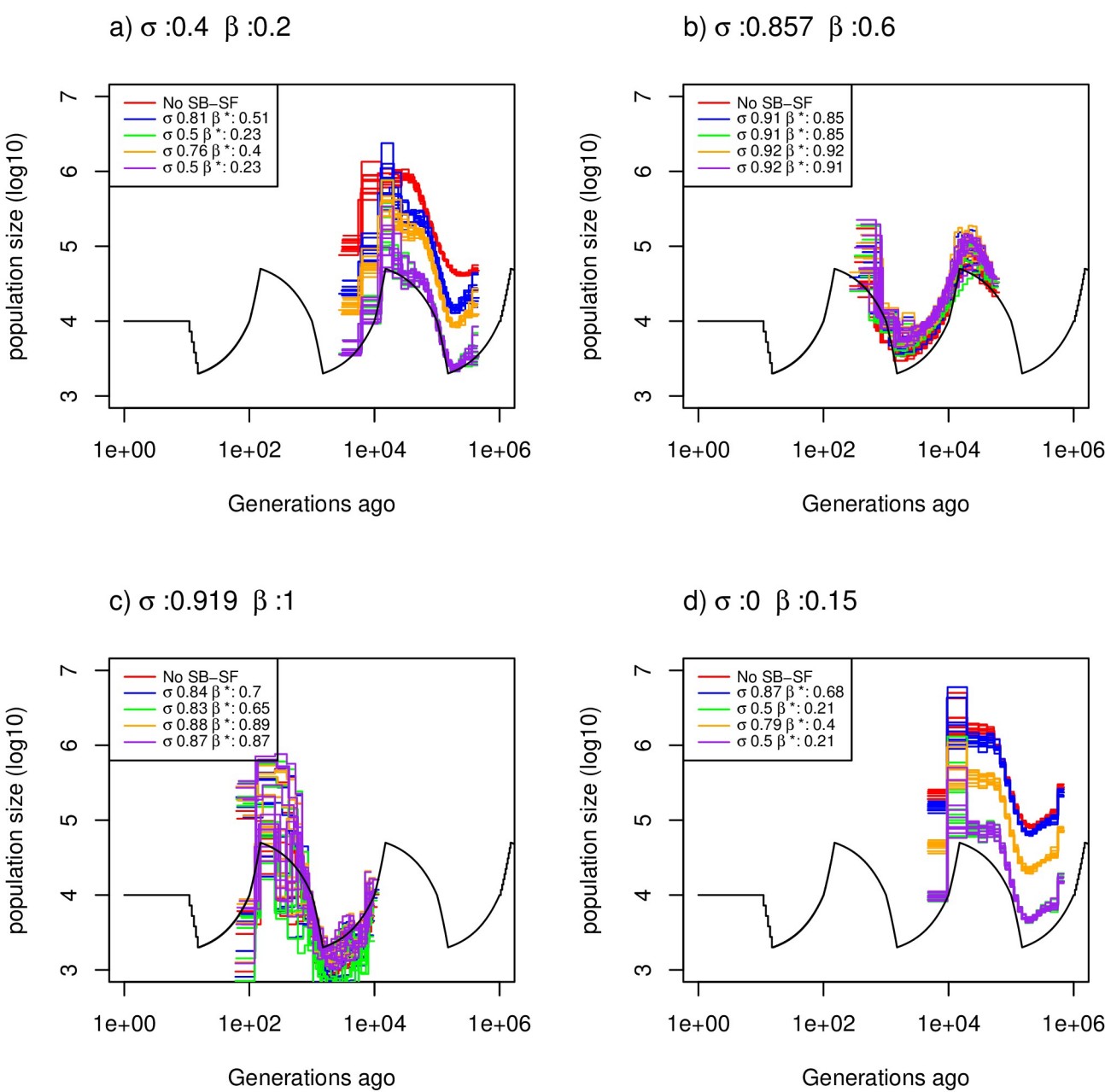

**Fig 4. Estimated demographic history with selfing and seed banking.** Demographic history estimated by eSMC for ten replicates using four simulated sequences of 10 Mb under a saw-tooth demographic scenario and four different combinations of germination ($\beta$) and self-fertilization ($\sigma$) rates but resulting in the same $\frac{\rho}{\theta}$. Mutation and recombination rates are set to $2.5 \times 10^{-8}$ per generation per bp, giving $\frac{r}{\mu} = 1$. The four combinations are: a) $\sigma = 0.4$ and $\beta = 0.25$, b) $\sigma = 0.75$ and $\beta = 0.6$, c) $\sigma = 0.85$ and $\beta = 1$ and d) $\sigma = 0$ and $\beta = 0.15$. Hence, for each scenario $\frac{\rho}{\theta} = 0.15$ For each combination of $\beta$ and $\sigma$, eSMC was launched with five different prior settings: ignoring seed-banks and self-fertilization (red), accounting for seed-banks and self-fertilization but without setting priors (blue), accounting for seed-banks and self-fertilization with a prior set only for the self-fertilization rate (green), only for the germination rate (orange) or for both (purple). $\sigma^*$ and $\beta^*$ respectively represent the estimated self-fertilization and germination rate.

## Inferring egg-banks and demography in *Daphnia pulex*

The inferred demographic history of a single population of *D. pulex* has a similar shape using eSMC and PSMC' (Fig 6). The demographic history estimated by PSMC' is shifted vertically

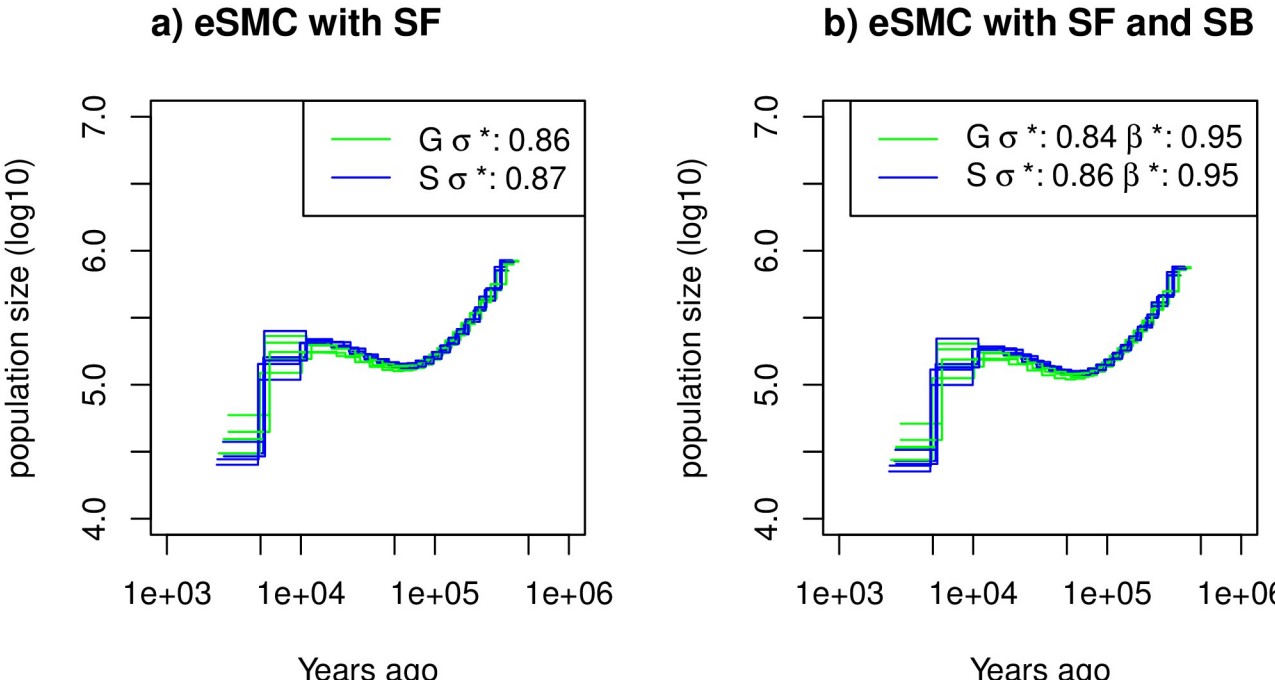

**Fig 5. Estimated demographic history of *Arabidopsis thalinana*.** Demographic history of two European (Sweden (S, blue) and German (G, green)) populations of *A. thaliana* estimated using eSMC: a) accounting only for selfing ($\sigma$ is a variable and $\beta = 1$) and b) accounting simultaneously for selfing and seed-banking ($\sigma$ bounded between 0.5 and 0.99 and $\beta$ bounded between 0.5 and 1). Mutation rate is set to $7 \times 10^{-9}$ per generation per bp and recombination respectively set for chromosome 1 to 5 to $3.4 \times 10^{-8}$, $3.6 \times 10^{-8}$, $3.5 \times 10^{-8}$, $3.8 \times 10^{-8}$, $3.6 \times 10^{-8}$) per generation per bp. $\sigma^*$ and $\beta^*$ respectively represent the estimated self-fertilization and germination rates.

compared to eSMC since dormancy is ignored. The effective population size is hence overestimated compared to eSMC. We fix the self-fertilization rate at $\sigma = 0$ because during sexual cycles although *D. pulex* in principal could self-fertilize via intraclonal matings during sexual cycles, these matings are rare and it has been shown that selfing is negligible in this case [4]. *D. pulex* reproduces via cyclical parthenogenesis, i.e. alternating phases of ameiotic parthenogenesis (or more exactly abortive meiosis with no or very little recombination [47, 48]) and sexual reproduction. Hence, the inferred mean generation time before the hatching of dormant eggs produced by sexual reproduction depends on the number of parthenogenetic cycles that occur on average per year since mutation can occur during ameiotic parthenogenesis but recombination is very unlikely. Here one generation is considered to be of one cycle of asexual or sexual reproduction, several generations taking place in a single year. For this specific population, a maximum of five parthenogenetic cycles before sexual reproduction is assumed [4]. It is important to note that the number of parthenogenetic cycles can affect the ratio $\frac{\rho}{\theta}$. Therefore we tested the effect of the value of the average number of parthenogenetic cycles on the estimation of the germination rate. Independently of the number of parthenogenetic cycles, eSMC always detects dormancy, with $\beta < 0.3$. The average dormancy can therefore be bounded between 3 and 18 generations, revealing the existence of at least moderate dormancy in this species.

## Discussion

The existing statistical inference methods based on full genome polymorphism data estimate the past demographic history under the assumptions of a model that is violated in many

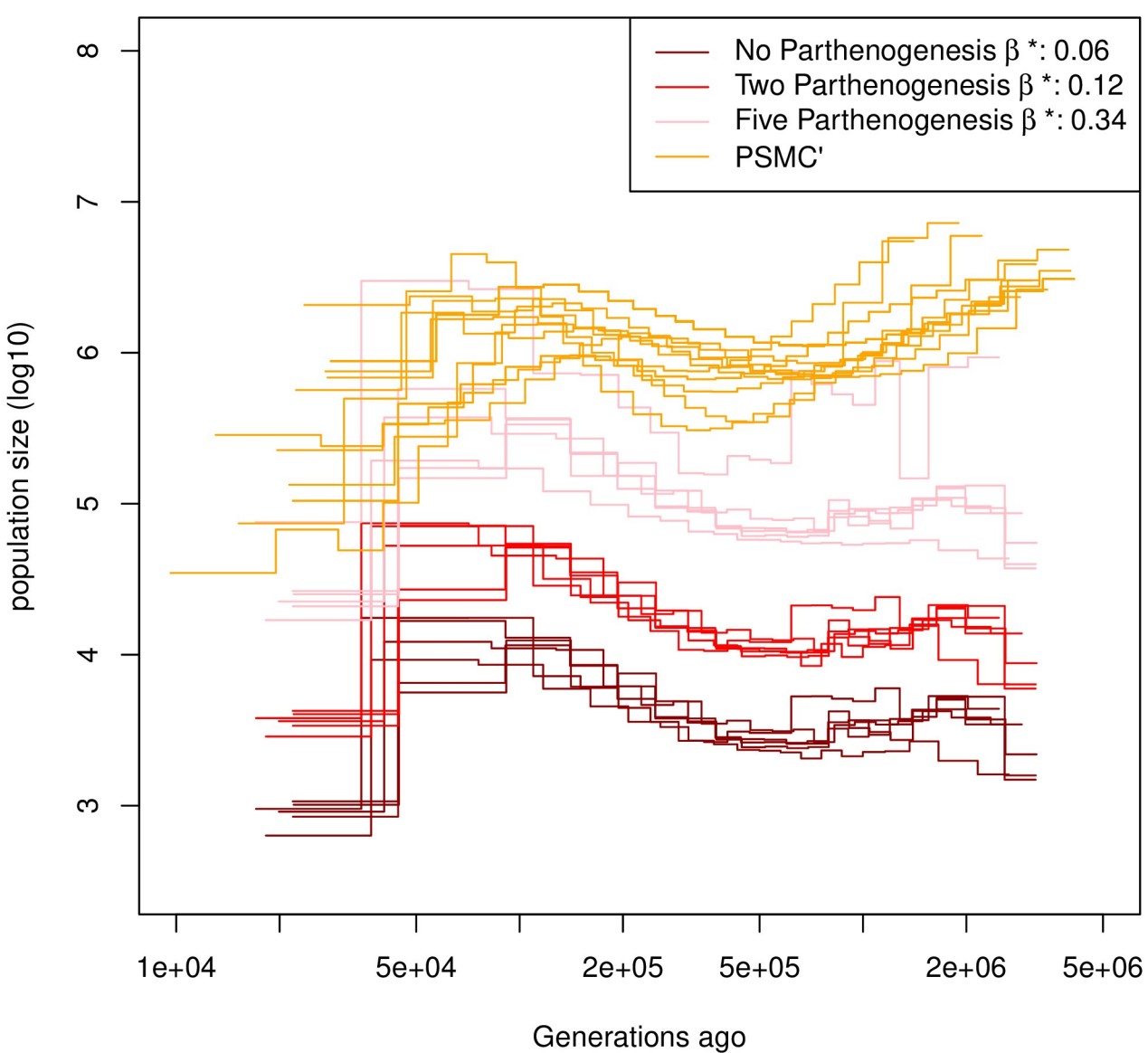

**Fig 6. Estimated demographic history of *Daphnia pulex*.** Demographic history estimated by eSMC on six individuals of *D. pulex* accounting for egg-banks (*β* is a variable and *σ* = 0). Different assumptions concerning the number of parthenogenetic cycles before the production of the dormant egg are made: Five cycles (pink), two cycles (red) and no parthenogenesis (dark red). A subset of demographic history estimated by PSMC' are ploted in orange. Mutation and recombination rates are respectively set to $4.33 \times 10^{-9}$ and $\frac{8 \times 10^{-8}}{n_p}$ per generation per bp, where $n_p$ is the number of reproductive cycles per year, parthenogenetic and sexual.

species. Here, we develop a method where ecological and life history traits can not only be accounted for, but can also be inferred from sequence data along with the past demography. Ecology and life history traits can affect $\rho$ and $\theta$ differently and our HMM can detect these differences through the estimation of $\frac{\rho}{\theta}$. However, this implies that some knowledge of the molecular ratio of recombination over mutation ($\frac{r}{\mu}$) is required. We demonstrate the capacity of our method to accurately recover the germination rate (and therefore the presence and strength of dormancy), though simulated results show that the violation of the infinite site assumption can lead to it being slightly underestimated. In a similar way, our model can also retrieve the

self-fertilization rate and we show that for high rates ($\sigma \geq 0.9$), more data are required to compensate for the variance observed and increase the accuracy of the estimation. Finally, our model cannot disentangle the genomic signatures of self-fertilization and seed-banks due to non-identifiability. The simultaneous estimation should thus be avoided, and only be performed if a priori knowledge on the presence seed-banks (or a dormant stage) as well as reproductive mode is available [21, 23].

We demonstrate that, even in the absence of self-fertilization and dormancy, the accuracy of eSMC in inferring demographic history is similar to, and in several cases better than, other methods that are widely used. This is due to several factors: 1) eSMC runs all pairwise analysis, increasing the amount of available information and the capacity to bound estimated values, 2) MSMC infers demographic history in a more recent time window, and requires more data for estimations with lower variance, and 3) Although MSMC2 also performs all pairwise analysis, the demographic history is estimated over a larger time window than any other tested methods, using the same amount of data, which increases the variance of the estimations. It is important to note that abrupt or sudden scenarios, such as bottlenecks, are difficult to infer, irrespective of the method used [49]. The effect of sharp (sudden) demographic events on allele frequencies can in fact be delayed for a small sample size, explaining why the demographic history of the bottleneck is smoothed [50]. Because our model can efficiently integrate information from several sequences and chromosomes, it gives good convergence properties with only a few relatively short sequences (a total of 10 Mb is sufficient, but each scaffold should be longer than 1 Mb). eSMC therefore allows an accurate estimation of demographic history with just one or two sequenced individuals. When using this method the quality of the genetic sequences should prevail over their quantity (in this, our method is similar to PSMC').

Throughout the paper we have highlighted two main ratios that are of great importance when using inference methods: the ratios $\frac{r}{\mu}$ and $\frac{\rho}{\theta}$, respectively the per site molecular and effective ratios of recombination over mutation rate. We have used the deviation between $\frac{r}{\mu}$ and $\frac{\rho}{\theta}$ to estimate the self-fertilization and/or germination rate. We also show that the demographic history can contribute to a departure of $\frac{\rho}{\theta}$ from $\frac{r}{\mu}$. Indeed, care must be taken concerning the initial value used for $\frac{\rho}{\theta}$: if the initial value is greater than one, the inferred demographic history will be flattened, regardless of the actual value of $\frac{\rho}{\theta}$. Furthermore, if the true value of $\frac{\rho}{\theta}$ is indeed greater than one, similar biases are expected, such as flattened demographic history. This observation, which is true irrespective of the presence of seed-banks or self-fertilization, is due to insufficient information to correctly reconstruct the local coalescent trees, as a high ratio $\frac{\rho}{\theta}$ implies that few SNPs are present between the recombination spots on the genome. We highlight that the importance of $\frac{\rho}{\theta}$ for inference was mentioned in [10], but has largely been ignored in the literature on SMC-based methods, despite the fact that a ratio of $\frac{\rho}{\theta}$ greater than one significantly alters the accuracy of inference.

When applying eSMC to sequences data of *A. thaliana*, we find evidence of strong self-fertilization with an estimated selfing rate around 0.87. However, this rate is slightly smaller than what is known empirically for this species, where the current rate of self-fertilization has been estimated at 0.99 [36]. There are three possible explanations for this discrepancy. First, *A. thaliana* most probably evolved from outcrossing to highly self-fertilizing less than 400 thousand years ago [46], whereas our demographic inference dates further in the past. Self-fertilization would therefore have appeared within the time window of the inferred demographic history. As a consequence, our estimate of self-fertilization (constant in time)

reflects the average effect of the varying the real self-fertilization rate within the time window. Second, the under-estimation may be due to limits of the self-fertilization model, which accounts only for homologous recombination events. Yet, other types of recombination or chromosomic re-arrangements do occur in genomes. These non-accounted for mechanisms could increase the signature of recombination leading to an underestimation of the self-fertilization rate. Third, we infer the self-fertilization rate of a single population in isolation (Germany or Sweden) while the past demography of *A. thaliana* consists of episodes of admixture, migration and recolonization from glacial refugia [3, 51], all of which are ignored in our model. The resulting complex population structure likely has an effect on our estimates (see discussion in [52]).

It has long been observed that many *Daphnia* species, including *D. pulex*, have resting egg-banks. The sequences analyzed using eSMC agree with this hypothesis, as we find strong evidence of dormancy. The inferred duration of dormancy greatly depends on the number of parthenogenetic generations between sexual reproduction events. Indeed, parthenogenetic cycles increase the number of mutations compared to recombination events. If we take two extreme scenarios for the specific sampled population (no parthenogenesis versus 5 generations of parthenogenesis [4]) we find a duration of dormancy between 3 and 18 generations. This is in agreement with empirical observations [21, 39], and confirms the major role of egg-banks in maintaining diversity in this species. The sequences used here originate from an ephemeral (*i.e.* non-permanent pond), and populations in such environments are expected to have both higher rates of sexual reproduction as well as longer-lived egg-banks [39]. It would therefore be interesting to test the existence of egg-banks and assess the germination rates in several *Daphnia* species and from different permanent and ephemeral water bodies. Our method presents a way forward to the detection of egg/seed/spore-banks of many invertebrates, plant and fungal species, as well as their past demographic history using sequence data (as experimental validation of dormancy is difficult to obtain [25, 31]).

Our method represents a first step for the integration of ecological traits in whole genome sequence analysis through the ratio $\frac{\rho}{\theta}$. We nevertheless advise caution when using our proposed, or other HMM methods (further advice and recommendations are found in [53]), for the inference of demography, as some assumptions may still be violated. For example we assume that mutations occur in the seed/egg-bank (a consequence of DNA damage) at the same rate as in the active population. While there is support in plants for this hypothesis [54, 55], we have no knowledge of supporting data in *Daphnia*. If mutations occur in seeds/eggs at a slower rate, we predict the estimation of dormancy to be a conservative lower bound, meaning that the seed/egg-bank is actually longer lived (*e.g.* [25]). Note finally, that all results rely on the quality of the sequences used and of the reference genome assembly.

In conclusion, the presented method is the first, to our knowledge, allowing the joint estimation of life-history traits and past demographic history based on full genome data. It is specifically adapted to the many species presenting violations of the classic Wright-Fisher model, and can be used to study the evolution of seed/egg-banking as an important bet-hedging strategy with large consequences on the rate of genome evolution [31].

## Materials and methods

### ecological Sequentially Markovian coalescenct(eSMC)

The eSMC is a Hidden Markov Model along two haplotypes. It is an extension to the PSMC' algorithm [12]. It adds the possibility of taking seed-banks and self-fertilization into account

and simultaneously estimating their rates along with the demographic history. As in PSMC', we assume neutrality, an infinite site model and a piece-wise constant population size. To define our HMM we need to precisely define all the following objects: the signal (observed data), the hidden states (coalescence time), the emission probabilities (probabilities of observing the data conditional to the hidden states), transition probabilities (probabilities of jump from one hidden state to another) and the probabilities of the initial hidden states. The demonstrations of the results presented here can be found in S1 Appendix.

The signal (or observed data) depends on the hidden state and is a chain of 0s and 1s. To construct this signal, as in PSMC', two genome sequences are compared at each position; if, at a given position, the two nucleotides are the same on both sequences, this is indicated by a 0, otherwise by a 1. As is necessary in HMM, the number of hidden states (or the coalescence times) must be finite, which is achieved by discretizing time. Therefore, the hidden state at one position is $\alpha$ if the coalescence time between the two haplotypes at that position is between $T_\alpha$ and $T_{\alpha+1}$. Given the model parameters, we know the expected coalescence time (which is $\frac{(2-\sigma)}{2\beta^2}$), and we define $T_\alpha$ as:

$$T_\alpha = \frac{-(2-\sigma)\ln\left(1-\frac{\alpha}{k}\right)}{2\beta^2} \tag{4}$$

Here, $k$ is the number of hidden states and $\alpha$ is an integer value between 0 and $k-1$. $\sigma$ and $\beta$ are the self-fertilization and the germination rate, respectively.

The emission probability $P$ is the probability of observing the signal (chain of 0's and 1's) conditional to the hidden states (coalescence time). As in the PSMC' algorithm, we consider an infinite site model. The emission rate is therefore given by:

$$P(0|\alpha) = 1 - e^{-2\mu t_\alpha}$$
$$P(1|\alpha) = e^{-2\mu t_\alpha}, \tag{5}$$

Where $\mu$ is the mutation rate per base pair and $t_\alpha$ the expected coalescence time in interval $\alpha$. We find:

$$t_\alpha = \frac{T_\alpha - T_{\alpha+1}e^{-\Delta_\alpha\Lambda_\alpha}}{(1 - e^{-\Delta_\alpha\Lambda_\alpha})} + \frac{1}{\Lambda_\alpha} \tag{6}$$

With:

$$\Delta_\alpha = T_{\alpha+1} - T_\alpha \;,\; \Lambda_\alpha = \frac{2\beta^2}{(2-\sigma)\chi_\alpha} \;,\; \text{and } \chi_\alpha = N_\alpha/N_e \tag{7}$$

Where $\Delta_\alpha$ is the duration (in coalescence time) of interval $\alpha$, $\Lambda_\alpha$ is the coalescence rate in the time window $\alpha$, $N$ is the effective population size and $N_\alpha$ is the population size during the time interval $\alpha$. Using $N$ and $N_\alpha$, we can calculate $\chi_\alpha$ which represents the variation of population size over time. It is this value that is inferred by the model.

The transition probabilities are the probabilities of going from one hidden state to another. We find:

$$
p(\alpha|\gamma) = \begin{cases}
\dfrac{P_\gamma}{2t_\gamma}\left(\left(\displaystyle\sum_{\eta=1}^{\alpha-1}\dfrac{(1-e^{-2\Delta_\alpha\Lambda_\alpha})e^{-\int_{T_{\eta+1}}^{T_\alpha}2\Lambda_\nu d\nu}(1-e^{-\Delta_\eta 2\Lambda_\eta})}{2\Lambda_\eta}\right)\right. \\
\qquad\qquad \left.+\left(\Delta_\alpha-\dfrac{(1-e^{-\Delta_\alpha 2\Lambda_\alpha})}{2\Lambda_\alpha}\right)\right) \\
\qquad\qquad\qquad\qquad \alpha < \gamma \\[2em]
\dfrac{P_\gamma}{t_\gamma}\left(\displaystyle\sum_{\eta=1}^{\gamma-1}e^{-\int_{T_{\eta+1}}^{t_\gamma}2\Lambda_\nu d\nu}\dfrac{(1-e^{-2\Delta_\eta\Lambda_\eta})}{2\Lambda_\eta}\right. \\
\qquad\qquad \left.+\dfrac{(1-e^{2(T_\gamma-t_\gamma)\Lambda_\gamma})}{2\Lambda_\gamma}\right)e^{-\int_{t_\gamma}^{T_\alpha}\Lambda_\nu d\nu}(1-e^{-\Delta_\alpha\Lambda_\alpha}) \\
\qquad\qquad\qquad\qquad \alpha > \gamma \\[2em]
1-\left(\displaystyle\sum_{\alpha=0}^{\gamma-1}p(\alpha|\gamma)+\sum_{\alpha=\gamma+1}^{k}p(\alpha|\gamma)\right) \\
\qquad\qquad\qquad\qquad \alpha = \gamma
\end{cases}
\tag{8}
$$

Where $P_\gamma$ is the recombination probability between two base-pairs:

$$
P_\gamma = \left(1 - e^{-2rt_\gamma\frac{2\beta(1-\sigma)}{(2-\sigma)}}\right)
\tag{9}
$$

The initial probability corresponds to the first state probability. We assume this probability to be the equilibrium probability $q_o(\alpha)$ (probability of being in state $\alpha$ at the first position). We find:

$$
q_o(\alpha) = e^{\sum_{\eta=0}^{\alpha-1}-\Lambda_\eta\Delta_\eta}(1-e^{-\Delta_\alpha\Lambda_\alpha})
\tag{10}
$$

### Simulated (pseudo-observed) sequence data

Throughout this paper we use five different demographic scenarios: 1) constant population size, 2) expansion, 3) bottleneck and recovery, 4) decrease and 5) "saw-tooth" (a succession of expansions and decreases). These scenarios are simulated for different combinations of the self-fertilization rate ($0 \leq \sigma \leq 0.9$) and the germination rate ($0.1 \leq \beta \leq 1$). Different sequence lengths are tested, as are combinations of mutation and recombination rates. To simulate our data, we use a modified version of the coalescence simulation program scrm [56]. This modified version integrates seed-banking (or egg-banking) and self-fertilization. The simulator is available on our GitHub repository (https://github.com/TPPSellinger/escrm). All the command lines can be found in S2 Appendix. On all the simulated data, four different algorithms are used to estimate demographic history and recombination rate: our algorithm eSMC, which

we compare to PSMC', MSMC and MSMC2. PSMC', MSMC and MSMC2 are run with default parameter and 1 as initial value for $\frac{\rho}{\theta}$.

## Sequence data

We use 12 whole genome sequences (hence all five chromosomes) of European *A. thaliana* from the 1001 genome project [3, 51], six individuals sampled in Sweden (id: 5830, 5836, 5865, 6077, 6085 and 6087) and six from Germany (id: 7231, 7250, 7255, 7337, 7415 and 7419). Each individual is considered haploid because of very high levels of homozygosity [19]. We obtained polymorphism data (that is processed vcf files) from the authors of the study [19]. The mapping to the reference genome and SNP call was performed based on the pipeline in [19]. The mutation rate is set at $7 \times 10^{-9}$ per generation per bp [57] and the chromosome specific recombination rates are $3.4 \times 10^{-8}$, $3.6 \times 10^{-8}$, $3.5 \times 10^{-8}$, $3.8 \times 10^{-8}$, $3.6 \times 10^{-8}$ per generation per bp for chromosome 1 to 5 respectively [45]. We first run the four different algorithms to estimate the demographic history and recombination rate (ignoring self-fertilization and seed-banks for eSMC). Analysis are run per chromosome (represented by the different lines in the figures). We then analyse the data again with eSMC, first accounting only for self-fertilization ($\beta$ is fixed to 1 and $\sigma$ is estimated), and then accounting for both self-fertilization and seed-banks using reasonable priors ($0.5 \leq \beta \leq 1$ and $0.5 \leq \sigma \leq 1$).

To infer the demographic history and the dormancy rates of *D. pulex*, we use six whole genome sequences from [4](id: SRR5004865, SRR5004866, SRR5004867, SRR5004868, SRR5004869 and SRR5004872) which are available under the accession SAMN06005639 in the NCBI Sequence Read Archive (SRA). We used the reference genome assembly PA42 v3.0 which is available at the European Molecular Biology Laboratory (EMBL) nucleotide sequencing database under accession PRJEB14656 [58]. The raw data was first trimmed using bbtools to remove duplicates, trim adapters, remove synthetic artifacts, spike-ins and perform quality-trimming based on minimum read quality of 40. Then we mapped reads using bwa (default parameters) onto the reference genome [59]. We used Samtools to convert sam to bam files [60] and GATK to remove PCR duplicates and perform local realignment around indels [61]. We used freebayes to call the SNPs and vcftools for post-processing (filtering). The pipeline is available on the GitHub repository: https://github.com/TPPSellinger/Daphnia_pulex_data. Note that the reference genome consists of 1,822 scaffolds (average length of 85,849) and thus to avoid bias in the analyses, we only kept scaffolds above 1 Mb retaining only the 19 largest scaffolds. As the phasing quality could not be guaranteed, we analyze sequence data of each *D. pulex* individual separately. The mutation rate is set at $4.33 \times 10^{-9}$ per generation per bp [62] and the recombination rate at $8 \times 10^{-8}$ per event of sexual reproduction per bp [4, 63]. So as to account for the number of generations before sexual reproduction takes place, the recombination rate is re-scaled by $n_p$ which represents the total number of generations per year. If we consider $n_p = 5$, the recombination rate is scaled by 0.2 [4]. We also test how the number of parthenogenetic generations between sexual reproductive events could affect the quality of the inference, and rescale the recombination rate accordingly. The scenarios we test are: no parthenogenesis, two generations of parthenogenesis and five generations of parthenogenesis, therefore rescaling the recombination rate by 1, 0.5 and 0.2 respectively. The sequences of each individual are analyzed with PSMC' and eSMC only, as MSMC and MSMC2 require accurate and reliable phasing, which is not the case for these sequences. We then account for egg-banks using eSMC and imposing no priors on $\beta$ and setting $\sigma = 0$. The multihetsep files for *A. thaliana* and *D. pulex* analysis are available on the GitHub repository (https://github.com/TPPSellinger/Daphnia_pulex_data and https://github.com/TPPSellinger/Arabidopsis_thaliana_data).

## Supporting information

**S1 Appendix. Complete model description.** Full and detailed description of the model and it's implementation, including mathematical demonstrations.
(PDF)

**S2 Appendix. Command lines.** Contains all scrm command line to reproduce all tested scenario.
(PDF)

**S1 Data. Numerical data.** File containing all numerical data presented in this study. It contains all plotted results and all requirements to build summary statistics.
(ZIP)

**S1 Fig. Estimated demographic history in absence of selfing or seed banking using sequences of 10 Mb.** Estimated demographic history using four simulated sequences of 10 Mb under a saw-tooth demographic scenario with 10 replicates. Mutation and recombination rate are set to $2.5 \times 10^{-8}$ per generation per bp. Therefore $\frac{r}{\mu} = \frac{\rho}{\theta} = 1$. The simulated demographic history is represented in black. a) Demographic history estimated by eSMC (red). b) Demographic history estimated by MSMC (purple). c) Demographic history estimated by MSMC2 (blue). d) Demographic history estimated by PSMC' (orange).
(TIF)

**S2 Fig. Estimated demographic history in absence of selfing or seed banking using sequences of 10 Mb when all method have same discretization of the population size as eSMC).** Estimated demographic history using four simulated sequences of 10 Mb under a saw-tooth demographic scenario with 10 replicates. Mutation and recombination rate are set to $2.5 \times 10^{-8}$ per generation per bp. Therefore $\frac{r}{\mu} = \frac{\rho}{\theta} = 1$. The simulated demographic history is represented in black. a) Demographic history estimated by eSMC (red). b) Demographic history estimated by MSMC (purple). c) Demographic history estimated by MSMC2 (blue). d) Demographic history estimated by PSMC' (orange).
(TIF)

**S3 Fig. Estimated demographic history in absence of selfing or seed banking using sequences of 1 Mb.** Estimated demographic history using four simulated sequences of 1 Mb under a saw-tooth scenario with 10 replicates. Mutation and recombination rate are set to $2.5 \times 10^{-8}$ per generation per bp. Therefore $\frac{r}{\mu} = \frac{\rho}{\theta} = 1$. The simulated demographic history is represented in black. a) Demographic history estimated by eSMC (red). b) Demographic history estimated by MSMC (purple). c) Demographic history estimated by MSMC2 (blue). d) Demographic history estimated by PSMC' (orange).
(TIF)

**S4 Fig. Estimated demographic history using eSMC in four simple demographic scenarios.** Estimated demographic history using four simulated sequences of 10 Mb under 4 different demographic scenarios with 10 replicates. Mutation and recombination rate are set to $2.5 \times 10^{-8}$ per generation per bp. Therefore $\frac{r}{\mu} = \frac{\rho}{\theta} = 1$. The simulated demographic history is represented in black. a) Demographic history simulated under a constant population size. b) Demographic history simulated under a bottleneck. c) Demographic history simulated under an expansion. d) Demographic history simulated under a decrease. Demographic history estimated by eSMC is in red.
(TIF)

**S5 Fig. Estimated demographic history using PSMC' in four simple demographic scenarios.** Estimated demographic history using four simulated sequences of 10 Mb under 4 different demographic scenarios with 10 replicates. Mutation and recombination rate are set to $2.5 \times 10^{-8}$ per generation per bp. Therefore $\frac{r}{\mu} = \frac{\rho}{\theta} = 1$. The simulated demographic history is represented in black. a) Demographic history simulated under a constant population size. b) Demographic history simulated under a bottleneck. c) Demographic history simulated under an expansion. d) Demographic history simulated under a decrease. Demographic history estimated by PSMC' is in orange.
(TIF)

**S6 Fig. Estimated demographic history using MSMC in four simple demographic scenarios.** Estimated demographic history using four simulated sequences of 10 Mb under 4 different demographic scenarios with 10 replicates. Mutation and recombination rate are set to $2.5 \times 10^{-8}$ per generation per bp. Therefore $\frac{r}{\mu} = \frac{\rho}{\theta} = 1$. The simulated demographic history is represented in black. a) Demographic history simulated under a constant population size. b) Demographic history simulated under a bottleneck. c) Demographic history simulated under an expansion. d) Demographic history simulated under a decrease. Demographic history estimated by MSMC is in purple.
(TIF)

**S7 Fig. Estimated demographic history using MSMC2 in four simple demographic scenarios.** Estimated demographic history using four simulated sequences of 10 Mb under 4 different demographic scenarios with 10 replicates. Mutation and recombination rate are set to $2.5 \times 10^{-8}$ per generation per bp. Therefore $\frac{r}{\mu} = \frac{\rho}{\theta} = 1$. The simulated demographic history is represented in black. a) Demographic history simulated under a constant population size. b) Demographic history simulated under a bottleneck. c) Demographic history simulated under an expansion. d) Demographic history simulated under a decrease. Demographic history estimated by MSMC2 is in blue.
(TIF)

**S8 Fig. Estimated demographic history under $\frac{r}{\mu} = 5$.** Results are obtained by fixing recombination rate to real value. Estimated demographic history using four simulated sequences of 10 Mb under a saw-tooth scenario with 10 replicates. Mutation and recombination rate are set to $2.5 \times 10^{-8}$ and $1.25 \times 10^{-7}$ per generation per bp. Therefore $\frac{r}{\mu} = \frac{\rho}{\theta} = 5$. The simulated demographic history is represented in black. a) Demographic history estimated by eSMC (red). b) Demographic history estimated by MSMC (purple). c) Demographic history estimated by MSMC2 (blue). d) Demographic history estimated by PSMC' (orange).
(TIF)

**S9 Fig. Estimated demographic history under $\frac{r}{\mu} = 100$.** Results are obtained by fixing recombination rate to real value. Estimated demographic history using four simulated sequences of 10 Mb under a saw-tooth scenario with 10 replicates. Mutation and recombination rate are set to $2.5 \times 10^{-8}$ and $2.5 \times 10^{-6}$ per generation per bp. Therefore $\frac{r}{\mu} = \frac{\rho}{\theta} = 100$. The simulated demographic history is represented in black. a) Demographic history estimated by eSMC (red). b) Demographic history estimated by MSMC (purple). c) Demographic history estimated by MSMC2 (blue). d) Demographic history estimated by PSMC' (orange).
(TIF)

**S10 Fig. Estimated demographic history under $\frac{r}{\mu} = 5$ with initial value $\frac{r}{\mu} = 5$.** Results are obtained by estimating recombination rate with initial value equal to mutation rate ($\frac{\rho}{\theta} = 5$).

Estimated demographic history using four simulated sequences of 10 Mb under a saw-tooth scenario with 10 replicates. Mutation and recombination rate are set to $2.5 \times 10^{-8}$ and $1.25 \times 10^{-7}$ per generation per bp. Therefore $\frac{r}{\mu} = \frac{\rho}{\theta} = 5$. The simulated demographic history is represented in black. a) Demographic history estimated by eSMC (red). b) Demographic history estimated by MSMC (purple). c) Demographic history estimated by MSMC2 (blue). d) Demographic history estimated by PSMC' (orange).
(TIF)

**S11 Fig. Estimated demographic history under $\frac{r}{\mu} = 5$ with initial value $\frac{r}{\mu} = 1$.** Results are obtained by estimating recombination rate with initial value equal to mutation rate ($\frac{\rho}{\theta} = 1$). Estimated demographic history using four simulated sequences of 10 Mb under a saw-tooth scenario with 10 replicates. Mutation and recombination rate are set to $2.5 \times 10^{-8}$ and $1.25 \times 10^{-7}$ per generation per bp. Therefore $\frac{r}{\mu} = \frac{\rho}{\theta} = 5$. The simulated demographic history is represented in black. a) Demographic history estimated by eSMC (red). b) Demographic history estimated by MSMC (purple). c) Demographic history estimated by MSMC2 (blue). d) Demographic history estimated by PSMC' (orange).
(TIF)

**S12 Fig. Estimated demographic history with seed banking and $\mu = 5 \times 10^{-9}$.** Estimated demographic history using four simulated sequences of 10 Mb and ten replicates under a saw-tooth demographic scenario (black). Simulation were done under four different germination rate $\beta$ (1,0.5,0.2 and 0.1). The mutation and recombination rates are set to $5 \times 10^{-9}$ per generation per bp. Therefore $\frac{r}{\mu} = 1$ and respectively $\frac{\rho}{\theta} = 1, \frac{\rho}{\theta} = 0.5, \frac{\rho}{\theta} = 0.2$ and $\frac{\rho}{\theta} = 0.1$. Estimated demographic history are represented for all tested germination rate, $\beta = 1$ (red), 0.5 (blue), 0.2 (green) and 0.1 (purple). The demographic history is estimated using a) eSMC where $\beta^*$ equal the estimated germination rate, b) MSMC, c) MSMC2 and d) PSMC'.
(TIF)

**S13 Fig. Estimated demographic history in four simple demographic scenarios with seed banking.** Estimated demographic history using four simulated sequences of 10 Mb under four different demographic scenarios with 10 replicates. Mutation and recombination rate are set to $2.5 \times 10^{-8}$ per generation per bp. Simulation were done under four different germination rate $\beta$. We have $\beta = 1$ (red), 0.5 (blue), 0.2 (green) and 0.1 (purple). Therefore $\frac{r}{\mu} = 1$ and we respectively have $\frac{\rho}{\theta} = 1, \frac{\rho}{\theta} = 0.5, \frac{\rho}{\theta} = 0.2$ and $\frac{\rho}{\theta} = 0.1$. The simulated demographic history is represented in black. a) Demographic history simulated under a constant population size. b) Demographic history simulated under a bottleneck. c) Demographic history simulated under an expansion. d) Demographic history simulated under a decrease. In addition we simulated data under four different germination rate $\beta$. $\beta^*$ equal the estimated germination rate.
(TIF)

**S14 Fig. Estimated demographic history in four simple demographic scenarios with seed banking where $\mu = 5 \times 10^{-9}$.** Estimated demographic history using four simulated sequences of 10 Mb under four different demographic scenarios with 10 replicates. Mutation and recombination rate are set to $5 \times 10^{-9}$ per generation per bp. Simulation were done under four different germination rate b. We have $\beta = 1$ (red), 0.5 (blue), 0.2 (green) and 0.1 (purple). Therefore $\frac{r}{\mu} = 1$ and we respectively have $\frac{\rho}{\theta} = 1, \frac{\rho}{\theta} = 0.5, \frac{\rho}{\theta} = 0.2$ and $\frac{\rho}{\theta} = 0.1$. The simulated demographic history is represented in black. a) Demographic history simulated under a constant population size. b) Demographic history simulated under a bottleneck. c) Demographic history simulated under an expansion. d) Demographic history simulated under a decrease. In addition we

simulated data under four different germination rate $\beta$. $\beta^*$ equal the estimated germination rate.
(TIF)

**S15 Fig. Estimated demographic history with selfing under $\frac{r}{\mu} = 5$.** Estimated demographic history using four simulated sequences of 10 Mb and ten replicates under a saw-tooth demographic scenario (black). Simulation were done under four different self-fertilization rate $\sigma$ (0,0.5,0.8 and 0.9). The mutation is set to $2.5 \times 10^{-8}$ and the recombination rate to $1.25 \times 10^{-7}$ per generation per bp. Therefore $\frac{r}{\mu} = 5$ and respectively $\frac{\rho}{\theta} = 5, \frac{\rho}{\theta} = 2.5, \frac{\rho}{\theta} = 1$ and $\frac{\rho}{\theta} = 0.5$. Estimated demographic history are represented for all tested self-fertilization, $\sigma = 1$ (red), 0.5 (blue), 0.2 (green) and 0.1 (purple). The demographic history is estimated using a) eSMC where $\sigma^*$ equals the estimated self-fertilization rate, b) MSMC, c) MSMC2 and d) PSMC'.
(TIF)

**S16 Fig. Estimated demographic history in four simple demographic scenarios with selfing.** Estimated demographic history using four simulated sequences of 10 Mb under four different demographic scenarios with 10 replicates. Mutation and recombination rate are set to $2.5 \times 10^{-8}$ per generation per bp. Simulation were done under four different self-fertilization rate $\sigma$ (0,0.5,0.8 and 0.9). Therefore $\frac{r}{\mu} = 1$ and respectively $\frac{\rho}{\theta} = 1, \frac{\rho}{\theta} = 0.5, \frac{\rho}{\theta} = 0.2$ and $\frac{\rho}{\theta} = 0.1$. The simulated demographic history is represented in black. a) Demographic history simulated under a constant population size. b) Demographic history simulated under a bottleneck. c) Demographic history simulated under an expansion. d) Demographic history simulated under a decrease. In addition we simulated data under four different self-fertilization rate $\sigma$. We have $\sigma = 0$ (red), 0.5 (blue), 0.8 (green) and 0.9 (purple). $\sigma^*$ equal the estimated self-fertilization rate.
(TIF)

**S17 Fig. Possible selfing and seed banking value where $\frac{r}{\mu} = 1$.** Possible estimated self-fertilization and germination rates because of confounding effect using four simulated sequences of 10 Mb under a saw-tooth demographic scenario and four different combinations of germination (b) and self-fertilization (s) rate but resulting in the same $\frac{\rho}{\theta} = 0.15$. Mutation rate is set to $2.5 \times 10^{-8}$ and recombination rate to $2.5 \times 10^{-8}$ per generation per bp. Therefore $\frac{r}{\mu} = 1$. The four combination are: a) $\sigma = 0.4$ and $\beta = 0.2$, b) $\sigma = 0.857$ and $\beta = 0.6$, c) $\sigma = 0.919$ and $\beta = 1$ and d) $\sigma = 0$ and $\beta = 0.15$. Hence, for each scenario $\frac{\rho}{\theta} = 0.15$ For each combination of $\beta$ and $\sigma$, eSMC was launched with five different prior settings: ignoring seed banks and self-fertilization (red), accounting for seed banks and self-fertilization but without setting priors (blue), accounting for seed banks and self-fertilization with a prior set only for the self-fertilization rate (green), only for the germination rate (orange) or for both (purple).
(TIF)

**S18 Fig. Estimated demographic history with selfing and seed banking where $\frac{r}{\mu} = 6.667$.** Demographic history estimated by eSMC for ten replicates using four simulated sequences of 10 Mb under a saw-tooth demographic scenario and four different combinations of germination (b) and self-fertilization (s) rate but resulting in the same $\frac{\rho}{\theta} = 1$. Mutation rate is set to $2.5 \times 10^{-8}$ and recombination rate to $1.667 \times 10^{-7}$ per generation per bp. Therefore $\frac{r}{\mu} = 6.67$. The four combination are: a) $\sigma = 0.4$ and $\beta = 0.25$, b) $\sigma = 0.75$ and $\beta = 0.6$, c) $\sigma = 0.85$ and $\beta = 1$ and d) $\sigma = 0$ and $\beta = 0.15$. Hence, for each scenario $\frac{\rho}{\theta} = 1$ For each combination of $\beta$ and $\sigma$, eSMC was launched with five different prior settings: ignoring seed banks and self-fertilization (red), accounting for seed banks and self-fertilization but without setting priors (blue), accounting for seed banks and self-fertilization with a prior set only for the self-fertilization

rate (green), only for the germination rate (orange) or for both (purple). $\sigma^*$ and $\beta^*$ respectively represent the estimated self-fertilization and germination rate.
(TIF)

**S19 Fig. Possible selfing and seed banking value where $\frac{r}{\mu} = 6.667$.** Possible estimated self-fertilization and germination rates because of confounding effect using four simulated sequences of 10 Mb under a saw-tooth demographic scenario and four different combinations of germination (b) and self-fertilization (s) rate but resulting in the same $\frac{\rho}{\theta} = 1$. Mutation rate is set to $2.5 \times 10^{-8}$ and recombination rate to $1.667 \times 10^{-7}$ per generation per bp. Therefore $\frac{r}{\mu} = 1$. The four combination are: a) $\sigma = 0.4$ and $\beta = 0.2$, b) $\sigma = 0.857$ and $\beta = 0.6$, c) $\sigma = 0.919$ and $\beta = 1$ and d) $\sigma = 0$ and $\beta = 0.15$. Hence, for each scenario $\frac{\rho}{\theta} = 1$ For each combination of $\beta$ and $\sigma$, eSMC was launched with five different prior settings: ignoring seed banks and self-fertilization (red), accounting for seed banks and self-fertilization but without setting priors (blue), accounting for seed banks and self-fertilization with a prior set only for the self-fertilization rate (green), only for the germination rate (orange) or for both (purple).
(TIF)

**S20 Fig. Estimated demographic history of *Arabidopsis thaliana* where selfing and seed banking is ignored.** Demographic history of two European (Sweden (blue) and German (green)) populations of *A. thaliana*. Mutation rate is set to $7 \times 10^{-9}$ per generation per bp and was use as prior for recombination rate. a) Demographic history estimated by eSMC without accounting self-fertilzation or dormancy. b) Demographic history estimated by MSMC. c) Demographic history estimated by MSMC2. d) Demographic history estimated by PSMC'.
(TIFF)

## Acknowledgments

We thank Gustavo Adolfo Silva Arias for his help in processing the data, Volker Hösel for his advice on Hidden Markov Models, and Michael Lynch and Zhiqiang Ye for the *Daphnia pulex* data.

## Author Contributions

**Conceptualization:** Thibaut Paul Patrick Sellinger.

**Data curation:** Thibaut Paul Patrick Sellinger.

**Formal analysis:** Thibaut Paul Patrick Sellinger.

**Funding acquisition:** Aurélien Tellier.

**Investigation:** Thibaut Paul Patrick Sellinger, Diala Abu Awad.

**Methodology:** Thibaut Paul Patrick Sellinger.

**Resources:** Thibaut Paul Patrick Sellinger.

**Software:** Thibaut Paul Patrick Sellinger.

**Supervision:** Aurélien Tellier.

**Validation:** Aurélien Tellier.

**Visualization:** Thibaut Paul Patrick Sellinger.

**Writing – original draft:** Thibaut Paul Patrick Sellinger, Diala Abu Awad.

**Writing – review & editing:** Thibaut Paul Patrick Sellinger, Diala Abu Awad, Markus Moest, Aurélien Tellier.

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
