## [Decision Letter · Decision Letter 0]

27 Aug 2019

Dear Dr Sellinger,

Thank you very much for submitting your Research Article entitled 'Inference of past demography, dormancy and self-fertilization rates from whole genome sequence data' to PLOS Genetics. Your manuscript was fully evaluated at the editorial level and by 3 independent peer reviewers. The reviewers appreciated the attention to an important problem, but raised some substantial concerns about the current manuscript. The main concern expressed by several reviewers focuses on the novelty of this approach and whether it represents a real improvement over the approaches it is clearly based upon. We would emphasize that the authors pay close attention to addressing this question, along with others raised in the reviews.

Based on the reviews, however, we will not be able to accept this version of the manuscript, but we would be willing to review again a much-revised version. We cannot, of course, promise publication at that time.

If you decide to revise the manuscript for further consideration at PLOS Genetics, please aim to resubmit within the next 60 days, unless it will take extra time to address the concerns of the reviewers, in which case we would appreciate an expected resubmission date by email to plosgenetics@plos.org.

[LINK]

We are sorry that we cannot be more positive about your manuscript at this stage. Please do not hesitate to contact us if you have any concerns or questions.

Yours sincerely,

Rodney Mauricio, Ph.D.

Associate Editor

PLOS Genetics

Gregory P. Copenhaver

Editor-in-Chief

PLOS Genetics

Reviewer's Responses to Questions

**Comments to the Authors:**

Reviewer #1: The inference of historical population size changes from genome

sequence data has received a lot of attention in recent

years. Sellinger et al. build on previous work in the Tellier group by

incorporating self-fertilization and seed dormancy into the

established HMM-framework. They show that their method, which they

call eSMC, infers population sizes in the presence of these

confounding factors more reliably than the three published methods

MSMC, MSMC2, and PSMC. The authors also find that dormancy has a

stronger effect on estimation than selfing.

The authors supply two accompanying programs: scrm for simulating data

under their model, and the R-package eSMC for estimation. The programs

run as described, though the test script hasn't finished yet after

over one hour and is producing a host of "unsuccessful convergence"

messages. It would therefore much improve the paper if the authors

supplied more extensive user documentation than their current README.

Apart from that, eSMC looks like a useful addition to the popgen

toolkit. My suggestions for further improving the paper concern again

mainly the computational side:

- p. 9, l 207ff: "our implementation in R is thus slightly slower, but

still allows whole genome sequence analysis of several individuals

in reasonable amount of time". This could be made more concrete: How

does the run time of eSMC vary as a function of the number of

polymorphisms, individuals, etc.?

- p. 17, l. 393f: The authors explain that the greater accuracy of

their model is due to running all pairwise analyses. This sounds

like a classical trade-off between accuracy and computational

effort. Again, a systematic study of run time as a function of

pertinent parameters would be helpful to users wishing to apply

eSMC.

- p. 45, Table 2: This contains only the averages of the results

returned by the various methods. Variances would also be useful for

establishing their relative merits.

- Minor: The authors repeatedly refer to "coalescent times"

(e.g. p. 7, l. 151) where they might prefer to write "coalescence

times" instead.

Reviewer #2: The paper presents a new method to infer demographic history and two important ecological parameters from genomic data, namely the self-fertilization rate and seed dormancy. The model is essentially a modified PSMC/PSMC' model, and the two additional parameters are important within plant population genetics and ecology. I think the key model and results are supported by evidence, and the authors have already provided many tests and simulations to assess the accuracy of their method and to compare with previous methods.

Before I outline some comments, as a general note: Providing the figures completely separate from the captions (and not even in the right order), and again completely separate from the main text makes life unnecessarily hard for reviewers! I think PLOS Genetics should encourage authors to submit nicely integrated documents to be sent to the reviewers instead of this puzzle work where one has to go back and forth between main text, figure captions and actual figures.

I have some major requests for clarification and comments:

- I generally don't fully understand how this model (eSMC) is different from MSMC2 or PSMC'. As far as I understand from the model description (section 2 in the main text) and SI 1, eSMC is _exactly_ a PSMC' model, but with modified coalescence and recombination rates. If that is the case, then MSMC2 with a freely inferred recombination rate, and freely inferred coalescence rates, should be exactly equivalent to eSMC after rescaling those coalescence and recombination rates, or do I miss something? I think it's totally fine to reimplement it and integrate the two additional parameters beta and sigma more tightly into the inference framework, but the reader should be fully aware of what the differences to previous methods are. If the only difference are i) a re-interpretation of the effective coalescence and recombination rates obtained by PSMC', and ii) the pairwise running-scheme over all haplotypes (which is already the case in MSMC2 I think), then that should be said more clearly.

- Given the above point, I don't understand why the results between eSMC and MSMC2 are so different in terms of variance of results. Why has eSMC so much less variance in recent times compared to MSMC2? Specifically the differences between Figure S3 (eSMC) and S4 (PSMC) are striking. I can think of no good reason why eSMC should have less variance than MSMC2. It runs on the same amount of data and has the same underlying model. Could this be a bug in MSMC2?

- I don't understand whether there is _any_ information that allows eSMC to separately and simultaneously estimate sigma and beta. The authors are a bit murky on that. In lines 167ff they say "We therefore have a potential confounding effect between self-fertilization and seed-banking when observing the recombination and mutation ratio". Here the word "potential" should be removed, as it looks to me mathematically impossible to infer them both simultaneously, unless I misunderstand something. And in line 171ff "seed-banking can theoretically be equivalent to self-fertilization with higher effective population size". If I understand correctly, it is not just "theoretically equivalent" but "fully equivalent", and it shouldn't sound as if this was just a mere possibility or "potential problem". The discussion in section 3.1.4 doesn't make it any clearer. For example, in Lines 323ff it says: "Without any prior knowledge it is not possible for eSMC to estimate to correct set of parameters." And further below, that estimates "can be slightly biased"... I don't understand. IS there, or ISN'T there information in the data that allows in principle to disentangle the two? If there isn't, then I would expect the results not to be "slightly biased", but completely uniformly distributed within the priors.

- Simulations: MSMC, MSMC2 and PSMC default parameters are designed for about 100 times as much data. With this little data as used here (1, 10 and 30 Mb), arguably the time discretisation should be adapted. For example, I would suggest that for PSMC and MSMC2 to use the exact same patterning as in eSMC to make it more comparable. With MSMC this is difficult, since the distribution of first coalescence times has a different window than the the distribution of pairwise coalescence times (as used in eSMC, MSMC2 and PSMC), but at least for PSMC and MSMC2 this should easily be possible. And for MSMC at least the total number of time segments should be made equal to the number of segments used in eSMC.

Minor comments:

- The main figures are all pretty poorly styled. There is a _lot_ of white space around the actual curves, which are squished into relatively tight log-spaces. I would recommend to zoom in a bit to focus on the actual estimates and curves.

- What exactly is meant by PSMC'? Specifically, what software was used? PSMC, MSMC2 (with two phased haplotypes) or MSMC (with two phased haplotypes)? Command lines should be listed somewhere, sorry if I overlooked them.

- Table 2: Given the centrality of the rho/mu estimates in this paper, I think a simple average is not enough here. I would prefer some scatter plots, or Violin/Boxplots type charts to indicate what the variability of estimates with each of the methods is.

- L 261ff and other places: I was unclear on whether the rho/mu ratio is generally kept free to be inferred, or whether it's fixed.

- Fig 2a: Is there a bug in the legend? The red curve should have beta=1, not 0, right? Same thing in Figure S9a I think.

- Fig S15 typo "Sweeden".

- Figure 6 has no discussion on why the curves look like they look. It is simply said "The inferred demographic history of a single population of D. pulex is qualitatively similar using eSMC and PSMC (Fig 6)". But the curves are shifted a lot (vertically). Please add at least a sentence to discuss this, otherwise the uninitiated reader will have trouble aligning that graphic with the statement of being "qualitatively similar".

- What does "parthenogenetic" mean, and what are "parthenogenetic cycles"?

Reviewer #3: Please see attachment.

**Have all data underlying the figures and results presented in the manuscript been provided?**

Reviewer #1: Yes

Reviewer #2: Yes

Reviewer #3: None

PLOS authors have the option to publish the peer review history of their article (what does this mean?). If published, this will include your full peer review and any attached files.

Reviewer #1: No

Reviewer #2: No

Reviewer #3: No

---

## [Decision Letter · Decision Letter 1]

5 Dec 2019

Dear Dr Sellinger,

Thank you very much for submitting your Research Article entitled 'Inference of past demography, dormancy and self-fertilization rates from whole genome sequence data' to PLOS Genetics. Your manuscript was fully evaluated at the editorial level and by three independent peer reviewers. The reviewers appreciated the attention to an important topic but identified some aspects of the manuscript that should be improved. You should pay close attention to Reviewer 3's comments. Since two reviewers recommend tightening the Discussion, you should make  all efforts to shorten that section as well.

We therefore ask you to modify the manuscript according to the review recommendations before we can consider your manuscript for acceptance. Your revisions should address the specific points made by each reviewer.

[LINK]

Yours sincerely,

Rodney Mauricio, Ph.D.

Associate Editor

PLOS Genetics

Gregory P. Copenhaver

Editor-in-Chief

PLOS Genetics

Reviewer's Responses to Questions

**Comments to the Authors:**

Reviewer #1: All my comments have been addressed.

Reviewer #2: I think the manuscript has improved and most of my issues have been addressed.

Just some minor points:

1) L 258: should reference Table 2, not Table 3.

2) Figure 3 is not referenced from the main text.

3) L 366 "[...], as there is an identifiability issue in the case of joint inference". I would say "[...], since the separate rates are non-identifiable" or something clearer.

4) L 439 "In its current version, our model cannot disentangle the genomic signatures of

self-fertilization and seed-banks". This sounds like this was a technical or implementation issue, but it is fundamental, and so I would remove "In its current version".

5) The discussion is very long, and I think it contains some redundancy. The paragraph starting from line 463 overlaps with the one from line 515, so perhaps things can be shortened a bit.

Reviewer #3: I thank the authors for their revisions. As best I can tell, the current state of the manuscript/this method is as follows:

1. eSMC is the same model as PSMC' (aka MSMC with sample size n=1 diploid; line 180).

2. The time discretization that eSMC uses is different from the defaults used by these methods (266ff). Note that it is possible to alter the default time discretization of PSMC/MSMC using the -p option, but the authors do not seem to have tried this.

3. Like MSMC2, eSMC uses a composite likelihood over all samples, as well as the correct Poisson likelihood for emission data, instead of a linear approximation which assumes that \\theta * N_e << 1. (269, 562ff).

4. eSMC optimizes over additional selfing and dormancy parameters during the M-step, and also incorporates parameter constraints.

5. eSMC operates on raw sequence data, instead of aggregating observations in window sizes of 100 or 1000.

Features 1-3 are not new. Feature 4 is, as noted by multiple reviewers, not necessary at all; one of these parameters can be backed out from the estimate of \\hat{\\rho} and the other cannot be identified. Feature 5 assumes that individual 100/1000bp blocks are nonrecombining. I would not expect it to have a major impact on inference unless the recombination rate is exceptionally high, which is not the case here.

All of these could be achieved by either altering the default command-line settings MSMC, or making minor coding modifications. But I appreciate that it is not reasonable to expect practitioners to do this, so a new implementation can be valuable.

Still, I think the most useful information contained in this paper is how to best parameterize {P,M}SMC for non-human data. In particular, setting the time discretization correctly can have a big impact on the quality of the estimates, as has been shown recently by other authors (Parag & Pybus, 2019). This seems to be the main source of improvement of their method over earlier methods -- it has a better default discretization for this particular type of data. In general, the authors have done a lot of simulation work exploring what works and does not work when running coalescent HMMs on non-human data. Those findings are mostly relegated to the supplemental. To me this is the primary contribution of this work, and should expanded to become the main focus of the manuscript, rather than the novelty of eSMC per se.

Finally, the discussion around identifiability remains unsatisfactory, and actually became less clear in revision as it is now a hodgepodge of the original (erroneous) assertions, and the reviewer comments which contradict them. A clear, unambiguous statement that the parameters cannot be jointly estimated is all that is needed here. I did not find the claimed "explicit guidelines for users when simultaneously estimating these two variables" anywhere. In my opinion, the only guideline necessary is that this should not be done. The authors seem to want to write this off as an issue of bias, but it is much worse than that. Biased estimators can be consistent, but unidentifiable models can never be consistently estimated, even in the presence of priors.

**Have all data underlying the figures and results presented in the manuscript been provided?**

Reviewer #1: None

Reviewer #2: Yes

Reviewer #3: None

PLOS authors have the option to publish the peer review history of their article (what does this mean?). If published, this will include your full peer review and any attached files.

Reviewer #1: No

Reviewer #2: No

Reviewer #3: No

---

## [Decision Letter · Decision Letter 2]

24 Feb 2020

Dear Dr Sellinger,

We are pleased to inform you that your manuscript entitled "Inference of past demography, dormancy and self-fertilization rates from whole genome sequence data" has been editorially accepted for publication in PLOS Genetics. Congratulations!

Yours sincerely,

Rodney Mauricio, Ph.D.

Associate Editor

PLOS Genetics

Gregory P. Copenhaver

Editor-in-Chief

PLOS Genetics

Comments from the reviewers (if applicable):

Reviewer's Responses to Questions

**Comments to the Authors:**

Reviewer #2: Looks all fine. Good to go from my point of view.

**Have all data underlying the figures and results presented in the manuscript been provided?**

Reviewer #2: Yes

PLOS authors have the option to publish the peer review history of their article (what does this mean?). If published, this will include your full peer review and any attached files.

Reviewer #2: No

**Data Deposition**

http://datadryad.org/submit?journalID=pgenetics&manu=PGENETICS-D-19-01160R2

**Press Queries**

---

## [Editor Report · Acceptance letter]

25 Mar 2020

PGENETICS-D-19-01160R2 

Inference of past demography, dormancy and self-fertilization rates from whole genome sequence data 

Dear Dr Sellinger, 

We are pleased to inform you that your manuscript entitled "Inference of past demography, dormancy and self-fertilization rates from whole genome sequence data" has been formally accepted for publication in PLOS Genetics! Your manuscript is now with our production department and you will be notified of the publication date in due course.

With kind regards,

Matt Lyles

PLOS Genetics

On behalf of:
